# Deaggregation of mutant *Plasmodium yoelii* de-ubiquitinase UBP1 alters MDR1 localization to confer multidrug resistance

Ruixue Xu [1,3], Lirong Lin[1,3], Zhiwei Jiao[1], Rui Liang[1], Yazhen Guo[1], Yixin Zhang[1], Xiaoxu Shang[1], Yuezhou Wang[1], Xu Wang[1], Luming Yao[1], Shengfa Liu[1], Xianming Deng [1], Jing Yuan [1] ✉, Xin-zhuan Su[2] ✉ & Jian Li [1] ✉

Mutations in a *Plasmodium* de-ubiquitinase UBP1 have been linked to anti-malarial drug resistance. However, the UBP1-mediated drug-resistant mechanism remains unknown. Through drug selection, genetic mapping, allelic exchange, and functional characterization, here we show that simultaneous mutations of two amino acids (I1560N and P2874T) in the *Plasmodium yoelii* UBP1 can mediate high-level resistance to mefloquine, lumefantrine, and piperaquine. Mechanistically, the double mutations are shown to impair UBP1 cytoplasmic aggregation and de-ubiquitinating activity, leading to increased ubiquitination levels and altered protein localization, from the parasite digestive vacuole to the plasma membrane, of the *P. yoelii* multidrug resistance transporter 1 (MDR1). The MDR1 on the plasma membrane enhances the efflux of substrates/drugs out of the parasite cytoplasm to confer multidrug resistance, which can be reversed by inhibition of MDR1 transport. This study reveals a previously unknown drug-resistant mechanism mediated by UBP1 through altered MDR1 localization and substrate transport direction in a mouse model, providing a new malaria treatment strategy.

Malaria is a devastating vector-borne disease causing an estimated 249 million clinical cases and 608,000 deaths in 2022[1]. Drug resistance remains one of the greatest challenges to malaria control and elimination. Deciphering the mode of action and mechanism of drug resistance and developing new antimalarial drugs are urgently needed. In *Plasmodium falciparum*, mutations and/or copy number variations of the multidrug resistance transporter gene 1 (*Pfmdr1*) have been associated with parasite response to mefloquine (MFQ), lumefantrine (LUM), dihydroartemisinin (DHA), piperaquine (PPQ), and chloroquine (CQ)[2–6]. *Plasmodium* MDR1 is a homolog of the mammalian P-glycoprotein (P-gp) that belongs to the ATP-binding cassette (ABC) transporter family and normally resides on the membrane of the parasite's digestive vacuole (DV)[7,8]. PfMDR1 was reported to transport drugs from the parasite cytosol into DV[7,9], in contrast to the

mammalian P-gp that is localized on the plasma membrane to efflux substrates across the cell membrane[10,11].

Mutations of V2697F and V2728F in a de-ubiquitinase (or ubiquitin carboxyl-terminal hydrolase 1, UBP1) have been linked to *Plasmodium chabaudi* responses to artesunate (ATS), artemisinin (ART), and/or CQ[12–14]. Introductions of equivalent mutation of PcUBP1 V2697F into *P. falciparum* and *Plasmodium berghei* increased ART resistant levels[15,16], and the introduction of the PcUBP1 V2728F equivalent mutation caused ART and CQ resistance in *P. berghei*, but not in *P. falciparum*[15,16]. PfUBP1 variants were associated with ART-delayed parasite clearance in *P. falciparum* isolates from Kenya and Thailand[17–19]. A PfUBP1 R3138H substitution was shown to confer ART resistance in a ring-stage survival assay in vitro[20]. However, the substrates of *Plasmodium* UBP1 and the mechanism of how the mutations in *Plasmodium* UBP1 confer drug

[1]State Key Laboratory of Cellular Stress Biology, School of Life Sciences, Faculty of Medicine and Life Sciences, Xiamen University, Xiamen, Fujian 361102, China. [2]Laboratory of Malaria and Vector Research, National Institute of Allergy and Infectious Diseases, National Institutes of Health, Rockville, MD 20850, USA. [3]These authors contributed equally: Ruixue Xu, Lirong Lin. ✉e-mail: yuanjing@xmu.edu.cn; xsu@niaid.nih.gov; jianli_204@xmu.edu.cn

resistance remain unknown. In this study, we identify two new mutations in *Plasmodium yoelii* UBP1 (I1560N and P2874T) and link the mutations to high-level MFQ resistance. Evidence is presented that *P. yoelii* MDR1 is a substrate of UBP1, and that these mutations perturb UBP1 cytoplasmic aggregation and activity to increase MDR1 ubiquitination level, which alters MDR1 trafficking from parasite DV to the plasma membrane. The *P. yoelii* MDR1 on the plasma membrane can efflux solutes out of parasite cytosol leading to multidrug resistance. Our study unravels a previously unknown drug-resistant mechanism mediated by two proteins connecting pathways of drug transport and protein ubiquitination in malaria parasites.

## Results

### Mutations in UBP1 linked to MFQ resistance

Two isogenic clones NSR and NSS, derived from *P. yoelii* NSM, were established after MFQ selective pressure (40 mg/kg, two cycles each for 5 days) in vivo and passages through *Anopheles stephensi* mosquitoes, respectively (Supplementary Fig. 1a, b). NSS grew faster than NSR without drug pressure, reaching 35%–60% parasitemia on Day 4 post-injection (*ip*) of $1 \times 10^5$ iRBCs compared to 7%–20% for the NSR parasite (Fig. 1a, Supplementary Fig. 1c). The parasitemia for NSR declined to ~5% after MFQ administration (40 mg/kg) but increased again when the drug treatment was stopped (Fig. 1a). The NSR MFQ-resistant phenotype was stable after freeze-thawing cycles (liquid N2) and serial blood passages through mice. In contrast, the NSS parasite did not survive the same MFQ treatment (Fig. 1a, Supplementary Fig. 1b, d). We next treated the parasites with different daily dosages of MFQ (0.1 mg/kg to 40 mg/kg) on Day 0 post-injection (inoculum of $1 \times 10^6$ iRBCs, iv) for 4 days. NSR could survive all the treatments, but NSS could only survive at 2.5 mg/kg or lower dosages (Fig. 1b). The NSR and NSS are therefore isogenic parasites with potential mutations in the NSR to confer high-level MFQ resistance.

To investigate the molecular mechanism underlying resistance to MFQ, we crossed the NSR parasite with a genetically distinct MFQ-sensitive parasite BY265 (Supplementary Fig. 1e) to identify MFQ-resistant gene(s) using linkage group selection as described

previously[21,22]. After performing eight independent crossing experiments, we obtained 35 uncloned progeny pools, with 25 selected with MFQ (20 mg/kg). DNA samples from the 25 MFQ treated and 10 non-treated progeny pools were genotyped with 190 polymorphic microsatellites (MS)[23] on 14 parasite chromosomes. All progeny from the untreated group carried BY265 alleles (Supplementary Data 1), indicating faster growth for BY265 without drug pressure. Interestingly, the frequency of the BY265 alleles were lower than 50% for several chromosomes or chromosomal segments from the MFQ-treated groups (Fig. 1c, Supplementary Data 1), particularly a deep "selection valley" of ~160 kb (chromosome position: 311,552–471,990) on chromosome 2 that is syntenic with chromosome 1 of *P. falciparum*[24]. The locus was defined by markers Py171-2 and Py1244 and crossovers in progenies 3#2-4 and 7#4-3 (Fig. 1d, e). Within the locus, the BY265 alleles for three consecutive MS markers (Py1184, Py1308, and Py2673) were completely absent under the MFQ treatment, suggesting gene(s) in the locus strongly linked to the MFQ-resistant phenotype.

Using the Illumina HiSeq sequencing platform, we obtained whole-genome sequences for the isogenic lines NSS and NSR with 83X and 129X averaged coverage, respectively. Only 10 nonsynonymous single nucleotide polymorphisms (SNPs) in five genes were detected genome-wide after careful comparison of the sequences (Supplementary Table 1), and no mutation or copy number variation was found in the orthologues of *Pfcrt*, *Pfmdr1*, *Pfmdr2*, *Pfk13*, *plasmepsin 2*, and *plasmepsin 3* that have been implicated in various drug resistances. Among the nonsynonymous SNPs, two were in a gene encoding the UBP1 (PY17X_0210200) within the chromosome 2 selection valley, resulting in Ile→Asn (I→N, codon #1560) and Pro→Thr (P→T, codon #2874) amino acid substitutions with the NSR carrying the NT alleles and NSS having the IP alleles (Supplementary Table 1).

The entire *ubp1* gene from the NSS, NSR, and BY265 parasites was sequenced using Sanger sequencing and compared with those from other parasite strains or species downloaded from public databases (Supplementary Figs. 2a, b, 3). The wild-type IP alleles were conserved in all the MFQ-sensitive parasites, including *P. falciparum*, *P. chabaudi*, and *P. beghei* (Fig. 1f). The NT mutations were located within a

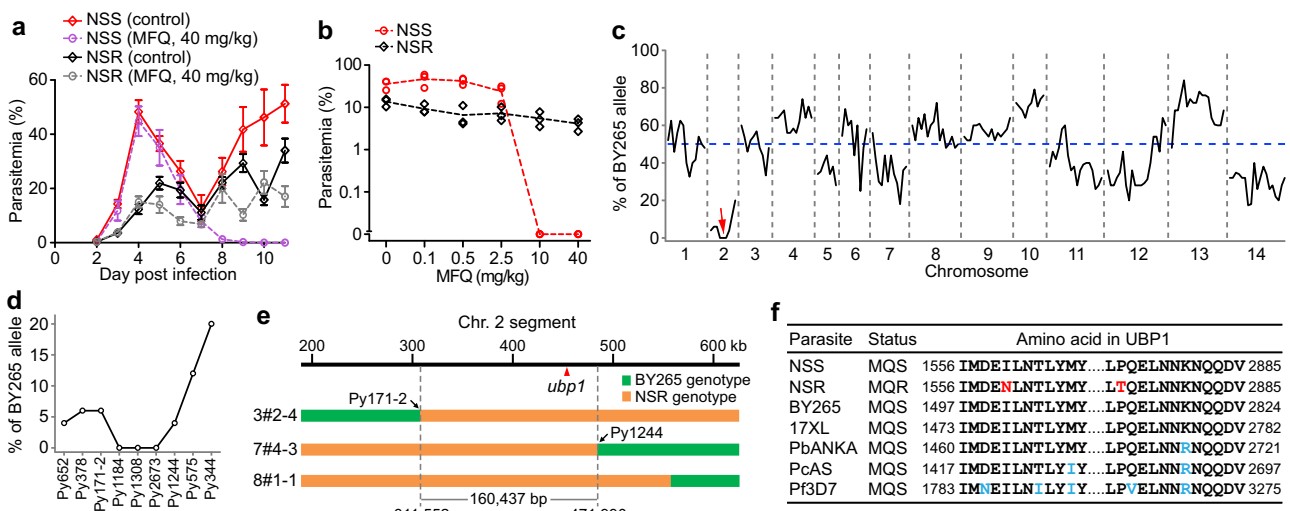

**Fig. 1 | Identification of UBP1 mutations linked to mefloquine (MFQ) resistance. a** Parasitemia from Balb/c mice infected with *P. yoelii* NSS and NSR with or without 40 mg/kg MFQ treatment for 4 days from day 4 post-infection. Mean ± SEM from three mice in the NSS group, and four mice in the NSR group. **b** Parasitemia from ICR mice infected with NSS and NSR treated with different dosages of MFQ, three mice for each group. **c** Frequencies of BY265 alleles of genome-wide microsatellite markers in MFQ selected progeny pools from genetic crosses between *P. yoelii* BY265 (MFQ sensitive) and NSR (MFQ resistant). The red arrow indicates a selection valley on chromosome 2. **d** An expanded chromosome 2 selection valley showing BY265 allele ratios of specific microsatellite markers. **e** A segment within the selection locus showing the position of a candidate gene *ubp1* linked to MFQ resistance and crossovers in three progenies flanking the locus. **f** Alignments of partial UBP1 sequences from *P. berghei* ANKA (PbANKA), *P. chabaudi* AS (PcAS), *P. falciparum* 3D7 (Pf3D7), and four *P. yoelii* lines (NSS, NSR, BY265, and 17XL). MQS, mefloquine sensitive; MQR, mefloquine resistance.

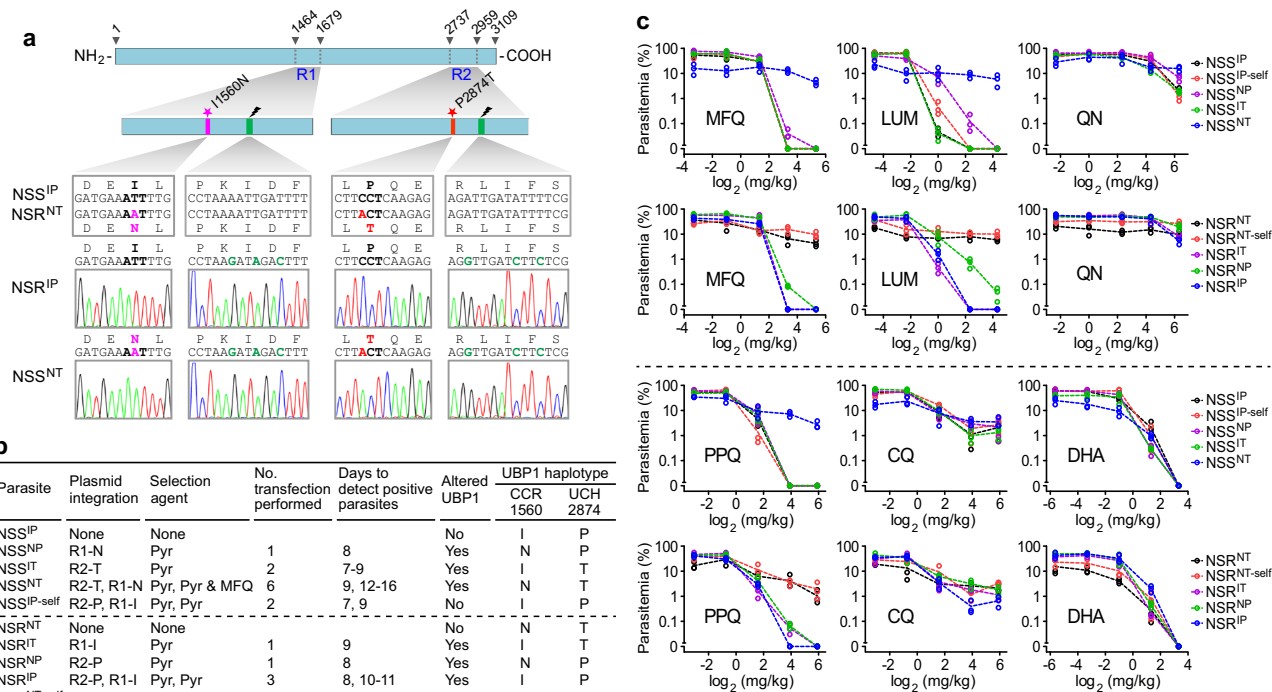

**Fig. 2 | Allelic replacements of UBP1 mutations between *P. yoelii* NSS and NSR alter parasite sensitivity to multiple drugs. a** Schematic showing UBP1 structure and the two regions being replaced using CRISPR/Cas9 editing method. The numbers on the top of the protein bar are amino acid positions indicating the start and end positions as well as the replacement regions. Partial nucleotide and amino acid sequences of nonsynonymous mutation (pink or red) between NSS and NSR, Cas9-gRNA targeting site (black arrowheads), and the silent mutation sites (green) to mutate the Cas9 cleavage site are presented within the regions. The chromatograms of DNA sequences confirming the exchanged nucleotides in the double cross-replacement clones are also shown. See also Supplementary Fig. 5. **b** Summary of allelic exchange experiments and haplotypes of parasite clones with *ubp1* modifications. CCR, conserved central region domain; UCH, ubiquitin carboxyl-terminal hydrolase domain. **c** In vivo antimalarial drug responses of *ubp1*-modified and parental parasite lines. MFQ mefloquine hydrochloride, LUM lumefantrine, QN quinine hydrochloride dehydrate, PPQ piperaquine phosphate, CQ chloroquine diphosphate, DHA dihydroartemisinin. Three mice were used for each dosage in each group; drug assays were as described in Methods.

conserved central region (CCR) and ubiquitin carboxyl-terminal hydrolase (UCH) domains (Supplementary Fig. 3), respectively. The CCR has some homology with the motifs III-IV of the glycosyltransferase (GTase) domain of bacterial penicillin-binding proteins (PBPs) (Supplementary Fig. 4a), as predicted using the Swiss-Model automated server (https://swissmodel.expasy.org/). The PBPs are bifunctional proteins containing GTase activity and transpeptidase (TPase) activity for peptidoglycan synthesis in bacteria including *Escherichia coli* and *Aquifex aeolicus*[25]. However, the CCR region shared only ~30% similarity with the motifs III-IV in GTase domain of the bacterial PBPs. Additionally, the UBP1 'KKKIMDEILNTL' motif (motif IV of bacterial PBP1) is also homologous to an ATP-binding cassette subfamily G member 1-like isoform X1 of *Belonocnema kinseyi* (Supplementary Fig. 4b) and a DNA repair protein of *Campylobacter concisus* (Supplementary Fig. 4c). The function of the UBP1 CCR is uncertain, and further investigation is required. To our knowledge, the IP → NT substitutions were identified for the first time, and the CCR domain has not been characterized in malaria parasites previously.

## The co-occurrence of NT substitutions in UBP1 confers high-level resistance to MFQ, LMF, and PPQ

The *ubp1* was reported to be refractory to disruption in *P. falciparum*[20]. We also designed two sgRNAs to disrupt the *P. yoelii ubp1* gene using the CRISPR/Cas9 system but failed to delete the gene in either NSS or NSR parasite after three attempts (Supplementary Fig. 2c, d). The results support that *ubp1* is essential for asexual erythrocytic stages.

We next performed allelic exchanges of the two UBP1 mutations (I1560N and P2874T) between NSS and NSR lines to investigate the roles of the mutations in modulating drug response and parasite fitness. Two DNA fragments covering the mutation sites were amplified from the parasites: Fragment R1 was a 644 bp DNA having I1560N substitution, and fragment R2 was a 667 bp DNA harboring P2874T substitution (Fig. 2a, Supplementary Fig. 5a). Two plasmids each containing the R1 or R2 sequence from NSS or NSR were cross transfected sequentially to replace IP with NT in the NSS parasite and NT with IP in the NSR parasite. Self-replacements (IP replaced with IP in NSS and NT replaced with NT in NSR) were also performed to serve as transfection controls, generating eight allelic replacements with four haplotype combinations, namely NSS^IP-self, NSS^NP, NSS^IT, NSS^NT, NSR^NT-self, NSR^IT, NSR^NP, and NSR^IP (Fig. 2b, Supplementary Fig. 5a). Single replacement clonal parasites of NSS^NP, NSS^IT, NSR^IT, and NSR^NP, double replacement NSR^IP, self-replacement NSS^IP-self, and NSR^NT-self were obtained from one to three rounds of transfection and pyrimethamine (Pyr) selection, but not NSS^NT parasites. NSS^NT clones were finally obtained after additional selection with 20 mg/kg MFQ for 4 days (Fig. 2b, Supplementary Fig. 5b), suggesting a disadvantage in vivo growth and MFQ-resistant trait of the NSS^NT parasites. This result is consistent with the finding that NSR parasite grew slower than the NSS parasite without drug selection (Supplementary Fig. 1c).

In vivo, drug assays testing parasite responses to MFQ, LUM, QN, PPQ, CQ, and DHA of the allelic exchanged parasites with NSS background showed that introduction of a single mutation, NSS^NP or NSS^IT, had a slight increase in IC90 to MFQ, LUM, and PPQ, but not IC50 values (Supplementary Table 2) or no effect on parasite responses to high levels of MFQ, LUM, and PPQ. In contrast, NSS^NT parasites with double substitutions dramatically increased parasite survival after MFQ, LUM, and PPQ treatments (Fig. 2c). As expected, in the set of parasites with NSR background, both NSR parent and

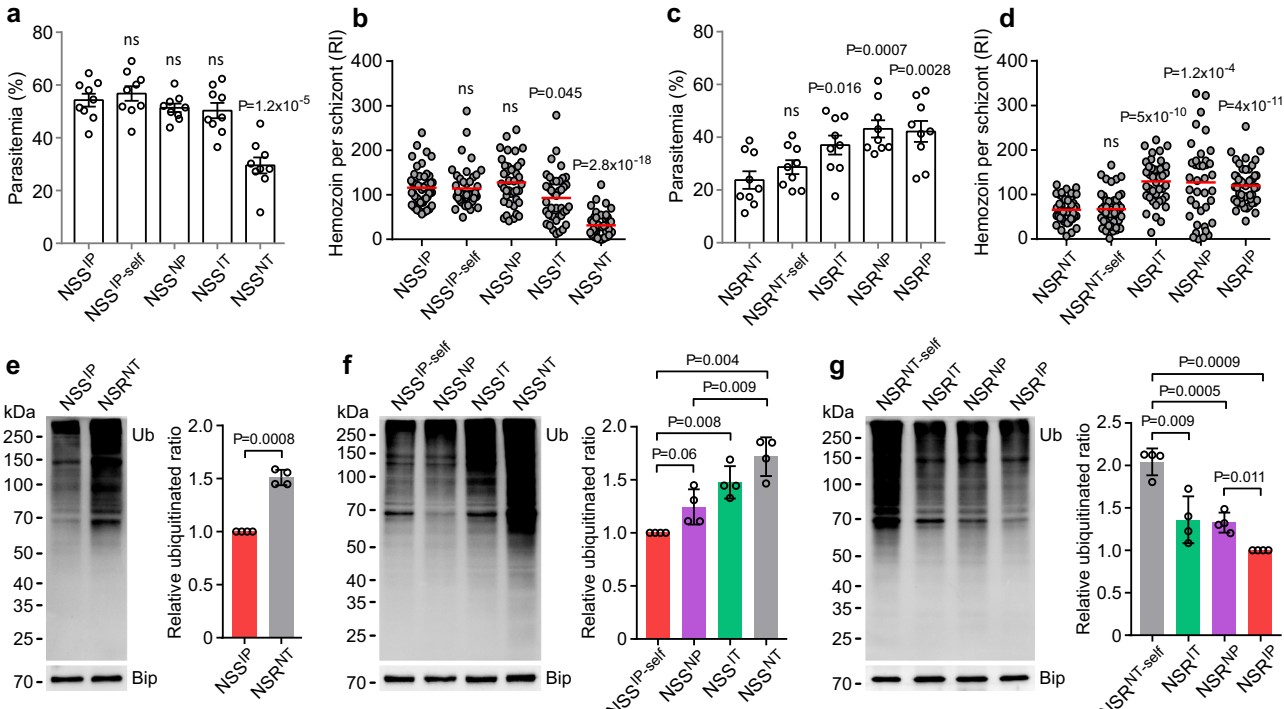

**Fig. 3 | UBP1 double mutations suppress parasite growth, hemozoin formation, and de-ubiquitination activity. a** Day-4 parasitemia (means ± SEM) from ICR mice infected with allelic-exchanged parasites in NSS background. $n = 9$ mice for each group; two-tailed $t$ test. **b** Relative intensity (RI) of hemozoin from schizonts of the parasites in (**a**). **c** Day-4 parasitemia (means ± SEM) from ICR mice infected with allelic-exchanged parasites in NSR background. $n = 9$ mice for each group; two-tailed $t$ test. **d** Quantification of hemozoin from schizonts of the parasites in (**c**). Hemozoin production inside iRBC was measured by reflection contrast polarized light microscopy in (**b**, **d**); data were obtained from 40 to 50 mature schizonts for each parasite; the horizontal lines show the mean values. Unpaired two-tailed $t$ tests were performed to compare the parental line NSS or NSR to each replacement in each plot of (**a–d**). **e–g** Left panels, Western blot showing protein ubiquitination in asexual blood stages of NSS, NSR, and allelic exchanged parasites. Ub, anti-ubiquitin antibodies; Bip, anti-Bip antibodies as loading control. Right panels, the bar graphs showing quantitative signals; scanned protein signals were normalized to that of Bip and plotted. Means ± SD of four independent experiments in (**e–g**); two-tailed $t$ test. **e** Global protein ubiquitination in asexual blood stages of NSS and NSR parasites. **f** Global protein ubiquitination in asexual blood stages of allelic-exchanged parasites in NSS background. **g** Global protein ubiquitination in asexual blood stages of allelic-exchanged parasites in NSR background.

NSR[NT-self] were resistant to MFQ, LUM, and PPQ, while single replacements (NSR[IT] and NSR[NP]) and double replacement (NSR[IP]) were sensitive to these antimalarials (Fig. 2c). These findings indicate that double mutations (1560 N plus 2874 T) are required for simultaneous resistance to high levels of MFQ, LUM, and PPQ. The NT mutations in the UBP1 had no significant effect on parasite responses to CQ, QN, and DHA (Fig. 2c).

## UBP1 double mutations impair parasite growth, hemozoin formation, and de-ubiquitinase activity

The difficulty in cloning NSS[NT] parasites suggests that parasites with the NT substitutions may impact the parasite's fitness. We, therefore, infected ICR mice with allelic exchanged parasites and counted early parasitemia (Day 4 post-infection) and measured hemozoin production of mature schizonts. The NSS[NT] parasite grew significantly slower and had lower hemozoin content per schizont than the other parasites with NSS background (Fig. 3a, b). Similarly, the parental NSR[NT] had significantly lower parasitemia and hemozoin counts than the allelic exchanged parasites (NSR[IT], NSR[NP], and NSR[IP]), but not the NSR[NT-self] (Fig. 3c, d). Collectively, these data show that the UBP1 NT substitutions significantly impact parasite growth and hemozoin formation. The reduced hemozoin formation may result in the accumulation of toxic free heme in the parasite cytosol.

To investigate whether the mutations in the UBP1 affect parasite protein ubiquitination, we performed a Western blot using a ubiquitin-specific antibody (P4D1, Cell Signaling Technology). Significantly higher levels of total ubiquitinated proteins were observed in the NSR[NT] than in the NSS[IP] parasite (Fig. 3e). Similarly, the allelic exchanges, either single or double replacements, also significantly altered total protein ubiquitination levels (except NSS[NP] had a $P = 0.06$) (Fig. 3f, g). These observations suggest that UBP1 is involved in de-ubiquitinating many unknown parasite proteins, and each substitution can impact its activity.

## NT haplotype alters UBP1 localization pattern and aggregation

To study the UBP1 protein localization and expression in different parasite stages, we tagged the *P. yoelii* UBP1 with a sextuple hemagglutinin (6HA) at N-terminus and a variable coding region between GDVKT and VQQSG residues in both NSS and NSR parasites using CRISPR/Cas9 method (Fig. 4a, Supplementary Fig. 6). Genes with correct tagging were identified by genotypic PCR (Supplementary Fig. 6d-a, d-b, d-c, d-d) and confirmed by DNA sequencing, generating HA-tagged parasites with NSS and NSR backgrounds, namely NSS[IP/ubp1::HAn] (N-terminal), NSS[IP/ubp1::HAm] (Middle), NSR[NT/ubp1::HAn], and NSR[NT/ubp1::HAm]. Western blotting analysis confirmed protein bands with expected ~375 kDa proteins (Fig. 4b); the expression levels of UBP1 had no statistically significant difference between parasites having IP and NT haplotypes (Fig. 4b).

The localization of endogenous fusion protein was examined using immunofluorescence assay (IFA). UBP1 was expressed as punctate structures in the cytoplasm of asexual blood-stage trophozoites and schizonts (Fig. 4c, Supplementary Fig. 7a); it was also expressed in gametocytes, gametes, zygotes, and ookinetes, but not in oocysts and sporozoites of the NSS[IP] (Supplementary Fig. 7a). Interestingly, UBP1 in

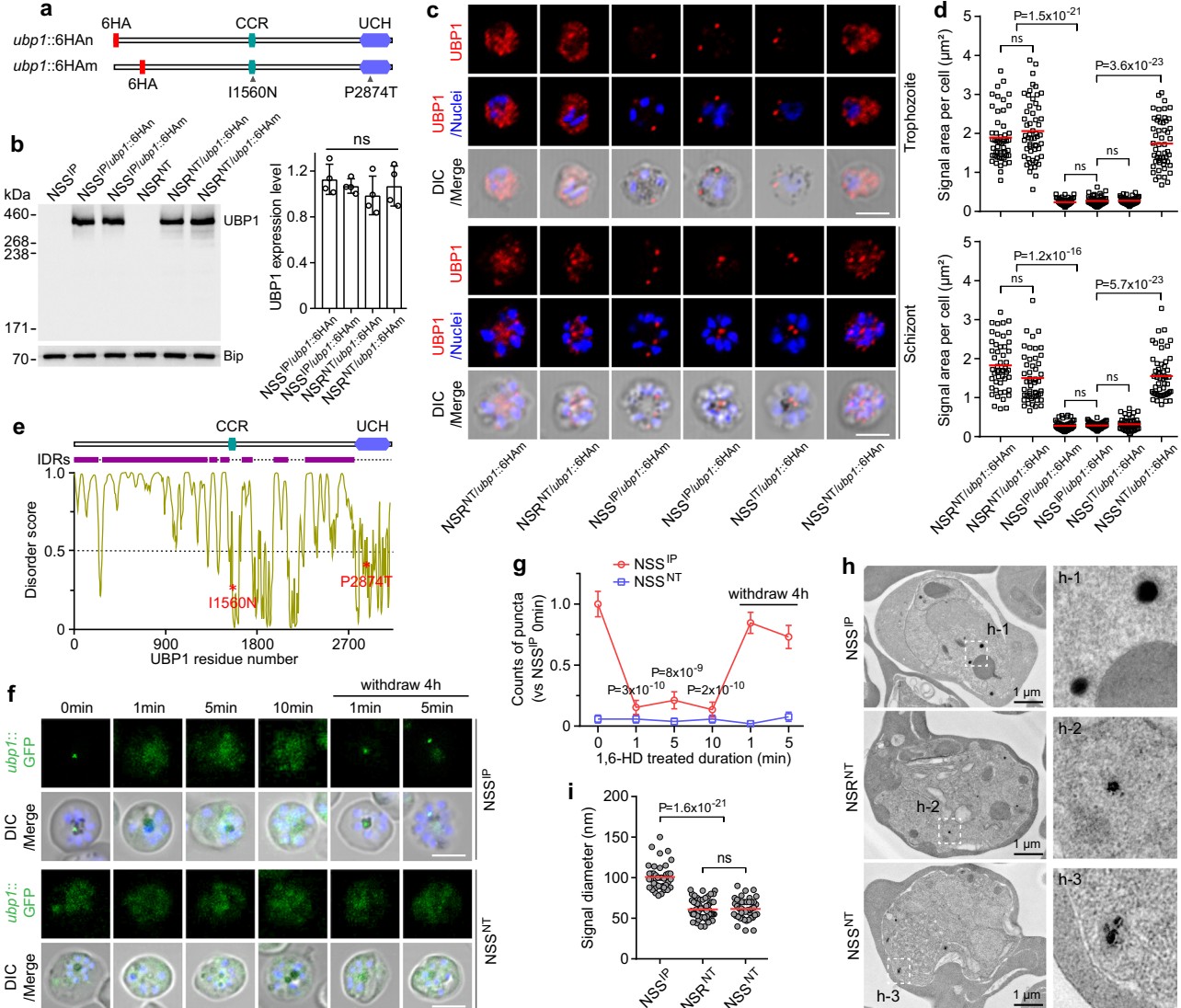

**Fig. 4 | UBP1 IP→NT substitutions change its cytosolic aggregation and distribution. a** Diagrams showing positions of amino acid substitutions and 6HA tagging in *P. yoelii* UBP1. CCR, conserved central region domain; UCH, ubiquitin carboxyl-terminal hydrolase domain. **b** Western blotting of UBP1 proteins in asexual blood stages of the modified parasites at using anti-HA antibodies. The bar graph shows the densitometric quantification of UBP1 expression; band densities were normalized to Bip. Means ± SD of four independent experiments; two-tailed *t* test. **c** IFA images of tagged UBP1 proteins in trophozoite, and schizont of modified parasites. Red, anti-HA; Blue, Hoechst 33342 stain; Scale bars, 5 μm. **d** Quantification of labeled UBP1 signals for trophozoite and schizont of the modified parasites in (**c**). *n* = 52, 55, 60, 55, 51, and 53 trophozoites were measured in NSR^NT/*ubp1*::HAm, NSR^NT/*ubp1*::HAn, NSS^IP/*ubp1*::HAm, NSS^IP/*ubp1*::HAn, NSS^IT/*ubp1*::HAn, and NSS^NT/*ubp1*::HAn, respectively; *n* = 50, 50, 51, 50, 50, and 54 schizonts were measured in NSR^NT/*ubp1*::HAm, NSR^NT/*ubp1*::HAn, NSS^IP/*ubp1*::HAm, NSS^IP/*ubp1*::HAn, NSS^IT/*ubp1*::HAn, and NSS^NT/*ubp1*::HAn, respectively. The horizontal lines show the mean values, two-tailed

*t* test. **e** A plot of intrinsically disordered regions (IDRs) in the UBP1 using PONDR (http://www.pondr.com/). A disorder score greater than 0.5 indicates IDRs. The red asterisks indicate the positions of the I1560N and P2874T mutations. **f** Live-cell images of the schizont stage of GFP-tagged UBP1 in NSS^IP and NSS^NT parasites that were treated with 10% 1,6-HD for different durations and after withdrawal of the compound. Scale bars, 5 μm. **g** Counts of fluorescent puncta within cytoplasm at each time point *versus* NSS^IP at 0 min in (**f**). Error bars represent standard errors from 60 cells for each time point per group; two-tailed *t* test. Representative for two independent experiments. **h** Representative images by transmission electron microscopy (TEM) of the *ubp1*::Apex2 trophozoites. The right panels indicate three representative Apex2 stained signals. **i** Diameter measurements of UBP1-Apex2 puncta from TEM images such as those in (**h**). *n* = 42, 59, 43 representative puncta images were analyzed in NSS^IP, NSR^NT, and NSS^NT, respectively. The horizontal lines show the mean values; two-tailed *t* test.

the NSR^NT or NSS^NT parasites was distributed more widely in the cytoplasm than NSS^IP and NSS^IT in the intra-erythrocytic stages (Fig. 4c, d). We engineered two additional parasite clones, NSS^IP/*ubp1*::GFP, and NSS^NT/*ubp1*::GFP, with *ubp1* being endogenously tagged with a GFP coding sequence at the N-terminus (Supplementary Fig. 6d-g, d-h). The GFP-tagged parasites further verified that NSS^NT had a more diffused expression of UBP1 than NSS^IP in the asexual blood stages (Supplementary Fig. 8a-c). These data show that IP→NT substitutions can affect UBP1 aggregation and distribution possibly by destabilizing

hydrophobic interaction after the introduction of the two hydrophilic residues.

The cytoplasmic UBP1 puncta in the NSS^IP parasite suggested that this structure could exist as a membrane-less biomolecular condensate formed by liquid-liquid phase separation (LLPS). One type of weak multivalent interaction in driving protein LLPS is mediated by intrinsically disordered regions (IDRs) or low complexity domains (LCD)[26,27]. Approximately 76% of the UBP1 sequence is predicted to be IDRs using PONDR (Fig. 4e). Hydrophobicity-dependent condensates can be

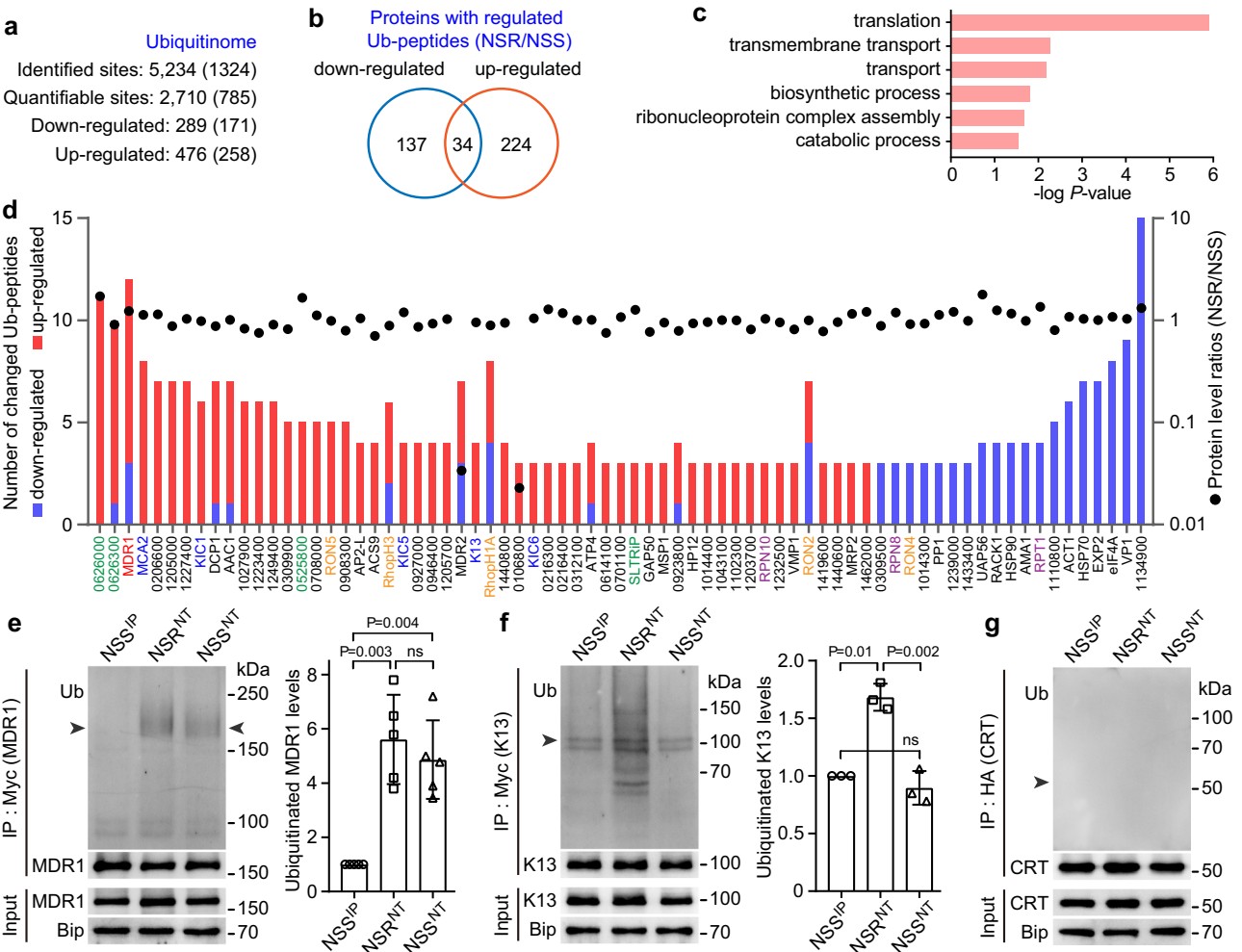

**Fig. 5 | Comparative profiling of global ubiquitination in the intra-erythrocytic stages of the NSS and NSR lines. a** The numbers of ubiquitination sites, quantifiable sites, and differential up-regulated and down-regulated sites of NSR versus NSS. The numbers in parentheses are the corresponding proteins. **b** Venn diagram of proteins with differential Ub-regulated sites. **c** Plot showing pathway enrichment analysis of the 395 proteins with regulated sites. **d** Proteins with the most ubiquitination regulated sites (≥3) of NSR/NSS; red and blue bars represent up-regulated and down-regulated sites, respectively. Protein level ratios of NSR/NSS are shown as a black dot for each protein. Detection of protein ubiquitination after immunoprecipitation (IP). Endogenous 4Myc-tagged MDR1 (**e**), 4Myc-tagged K13 (**f**), and 3HA-tagged CRT proteins (**g**) were extracted from asexual blood stages, followed by IP (anti-tag antibodies) and immunoblotting (IB, anti-ubiquitin antibodies) as indicated; Bip, anti-Bip antibodies as protein loading control. The bar graph shows the fold change of quantified ubiquitinated protein levels of MDR1 in (**e**) and K13 in (**f**), respectively. The arrowheads indicate the position of the predicted mass of protein. Data were presented as means ± SD; *n* = 5 and 3 independent experiments in (**e**) and (**f**), respectively; two-tailed *t* test.

disrupted by 1,6-hexanediol (1,6-HD) that interferes with the weak hydrophobic interactions between biomolecules[28,29]. Indeed, exposure of NSS[IP] parasites to 1,6-HD for 1 min, 5 min, and 10 min converted the UBP1 puncta into a diffused pattern, and ~75% of puncta was restored 4 h after 1,6-HD withdrawal (Fig. 4f, g). No detectable effect was observed in NSS[NT] parasites after 1.6-HD treatment (Fig. 4f, g). Thus, the substitutions of hydrophobic IP by hydrophilic NT amino acids likely contributed to the disruption of UBP1 condensate formation. However, we cannot rule out that 1,6-HD may also have off-target effects on other molecules in the cell. We next tagged the UBP1 with the engineered form of a soybean ascorbate peroxidase (APEX2) and observed the UBP1 condensates under transmission electron microscopy (TEM). Highly condensed spherical structures were observed in the NSS[IP] parasites (Fig. 4h). In contrast, dispersed and irregular structures were observed in NSR[NT] and NSS[NT] (Fig. 4h, i). These results suggest that the NT substitutions may perturb the UBP1 self-association and hence potentially alter its physiological functions as a de-ubiquitinase. Indeed, oligomerization is one of the mechanisms that can regulate DUB activities[30]. For example, human USP25 is

assembled into a homotetrameric quaternary complex that inhibits its enzymatic activity[31].

## *Plasmodium* UBP1 NT substitutions increase MDR1 ubiquitination

To identify the potential target proteins affected by the UBP1 NT mutations, we performed a global protein ubiquitination analysis of mixed blood stages from the isogenic lines NSS (drug sensitive) and NSR (drug resistant). To enrich Ub-modified peptides, we used an anti-diglycine-lysine antibody conjugated to agarose beads to capture peptides containing the ubiquitin remnant motif (K-ε-GG). Liquid chromatography-tandem mass spectrometry (LC-MS/MS) analysis of trypsin-digested proteins identified 5234 ubiquitinated peptides belonging to 1324 proteins. Compared with the NSS parasite, the NSR parasite had 476 peptides from 258 proteins with increased Ub signals and 289 peptides from 171 proteins with decreased Ub signals (differential signals ≥1.5-fold) (Fig. 5a, b; Supplementary Data 2 and 3). Gene Ontology (GO) analysis of the differentially ubiquitinated proteins showed significant enrichment of processes related to

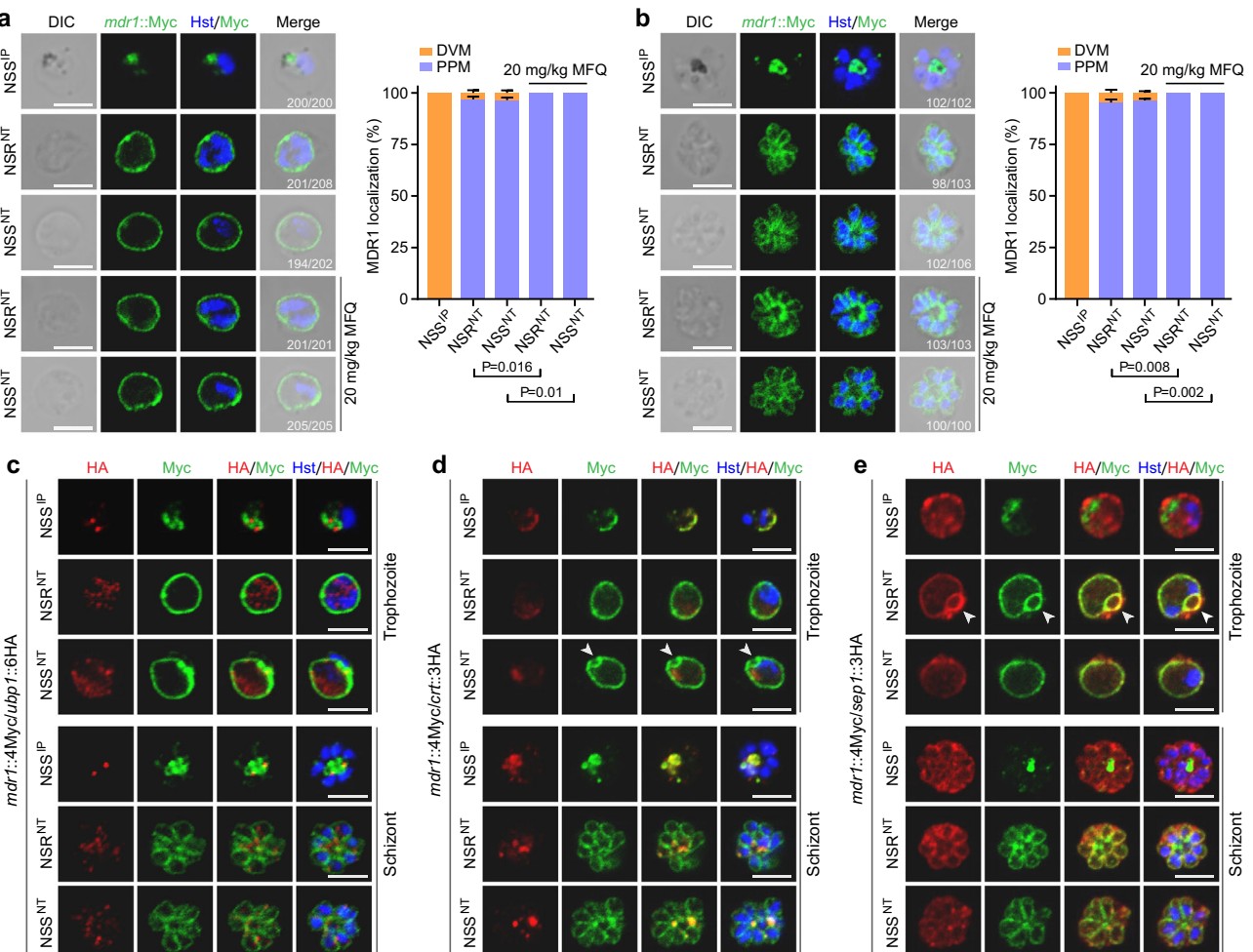

**Fig. 6 | UBP1 IP → NT substitutions alter MDR1 localization.** IFA analysis of parasite MDR1 expression and localization in trophozoite (**a**) and schizont (**b**). *mdr1*::4Myc parasites with UBP1 IP or NT haplotype were stained with anti-Myc antibodies. *x/y* in the figure is the number of cell displaying signal/the number of cell evaluated. The bar graphs in (**a**) and (**b**) show the percentages of MDR1 loca-lization at the digestive vacuole membrane (DVM) and parasite plasma membrane (PPM), respectively, in trophozoites or schizonts. Mean ± SD from four indepen-dent experiments, two-tailed *t* test. The dosages of mefloquine (MFQ) are as indicated. Blue, Hoechst 33342 stain. **c** Co-staining of MDR1 and UBP1 expressions in trophozoite and schizont of the *mdr1*::4Myc/*ubp1*::6HA parasites. Representative images from three independent experiments. **d** Co-staining of MDR1 and CRT expressions in trophozoite and schizont of the *mdr1*::4Myc/*crt*::3HA parasites. Representative images from three independent experiments. **e** Co-staining of MDR1 and SEP1 expressions in trophozoite and schizont of the *mdr1*::4Myc/ *sep1*::3HA parasites. Arrowheads indicate cytostome in (**d**, **e**). Scale bars, 5 µm. Representative images from three independent experiments.

translation, transmembrane transport, transport, biosynthesis, ribo-nucleoprotein complex assembly, and catabolism (Fig. 5c). The pro-teins had most up/down regulated sites (≥3) were shown on Fig. 5d. Ninety-three percent of the proteins in this list have an NSR/NSS ratio of protein expression level at 0.7–1.5, suggesting a lack of correlation between protein level and ubiquitination. Interestingly, NSR[NT] parasite had significantly up-regulated Ub-peptides from three tryptophan-rich proteins (PY17X_0626000, PY17X_0626300, PY17X_0525800), three K13 interaction candidate (KIC) proteins (KIC1, KIC5, and KIC6), K13, MCA2, and MDR1 compared to the levels of these peptides in the NSS[IP] parasite (Fig. 5d; Supplementary Data 2 and 3), suggesting that UBP1 potentially acts to de-ubiquitinate these proteins.

Among the proteins with increased Ub-peptides, *Plasmodium* MDR1 is known to play a role in parasite responses to multiple drugs. *P. yoelii* MDR1 had 42 ubiquitinated sites with nine sites having increased ubiquitination signals and three sites with decreased signals in the NSR[NT] parasite (Supplementary Fig. 9, Supplementary Data 3). To investigate the impact of the altered MDR1 ubiquitination on protein expression and trafficking, we tagged the endogenous MDR1 in the NSS[IP], NSR[NT], and NSS[NT] parasites with quadruple Myc epitope (4Myc)

at the C-terminus to obtain *mdr1*::4Myc parasites (Supplementary Figs. 6d-t, 4d-u, 4d-v). Because PfCRT and PfK13 contribute to parasite responses to CQ, DHA, and other drugs, we also tagged *P. yoelii* K13 with 4Myc at the N-terminus and CRT with 3HA at the C-terminus obtaining the *k13*::4Myc (Supplementary Fig. 6d-o, 6d-p, 6d-q) and *crt*::3HA (Supplementary Fig. 6d-l, 6d-m, 6d-n) parasites. Immuno-blotting analysis confirmed that NT substitutions in the UBP1 rendered higher levels of MDR1 ubiquitination (Fig. 5e), which was consistent with the data from the LC-MS/MS analysis. However, the NSR[NT] parasite also had a significantly higher level of ubiquitinated K13 than the NSS[IP] and NSS[NT] parasites (Fig. 5f), suggesting that other genetic determi-nants besides UBP1 contribute to the higher level of ubiquitinated K13 in the NSR[NT] parasite. No ubiquitinated protein could be detected for CRT in these parasites (Fig. 5g).

## UBP1 NT substitutions translocate MDR1 from the DV mem-brane to the plasma membrane

The *Plasmodium* MDR1 protein is normally localized on the parasite DV membrane (DVM)[7,8]. In the NSS[IP] trophozoites, MDR1 was expressed in hemozoin-containing DV scattered in the cytoplasm (Fig. 6a); in

schizonts, most of the MDR1 co-localized with large hemozoin (Fig. 6b). Similar pattern of MDR1 localization was observed in the NSS[NP] and NSS[IT] parasites (Supplementary Fig. 7b, c). Surprisingly, the MDR1 was mostly (more than 95% of cells) localized at the parasite plasma membrane (PPM) of trophozoites and schizonts in both NSR[NT] and NSS[NT] parasites (Fig. 6a, b). Furthermore, the percentage of cells that having PPM localized MDR1 increased to 100% under the MFQ pressure in the NSR[NT] and NSS[NT] parasites (Fig. 6a, b). One possibility for this observation was that MDR1 had not been adequately ubiquitinated in the small number of parasites. Alternatively, some parasites might revert to IP/NP/IT genotypes and express some MDR1 on DVM. Thus, the NT substitutions in UBP1 changed MDR1 trafficking from DVM to PPM, which may efflux drugs out of parasite cells.

To confirm the altered MDR1 protein localization, we genetically engineered three doubly tagged parasites (DTPs), *mdr1*::4Myc/ *ubp1*::6HA, *mdr1*::4Myc/*crt*::3HA and *mdr1*::4Myc/*sep1*::3HA from the *mdr1*::4Myc parasite (Supplementary Fig. 6d-w–6d-ae). Parasite proteins CRT and SEP1 are natively localized to DVM and parasitophorous vacuole membrane (PVM) that surrounds the parasite PPM[32,33], respectively. The localization patterns of MDR1 in the DTPs are consistent with the results from the single-tagged *mdr1*::4Myc parasites, e.g., results from DTPs confirmed that both UBP1 and MDR1 were expressed in blood stages, gametocytes, gametes, zygotes, and ookinetes, but not in oocysts and sporozoites of the NSS[IP] parasite (Fig. 6c, Supplementary Fig. 7a), suggesting similar temporal expression of the two proteins. However, MDR1 co-localized with the CRT in the NSS[IP] parasite, but not with UBP1 (Fig. 6c, d). Within the trophozoites having UBP1 NT haplotype, MDR1 was mostly detected on PPM as well as on the cytostome (Fig. 6d, e), an apparatus that mediated endocytic invagination for the uptake of host hemoglobin[34,35]. Additionally, the MDR1 signals in the UBP1 NT parasites appeared to be surrounded by the PVM protein SEP1 (Fig. 6e). The UBP1 NT substitutions did not change the protein localization of CRT and SEP1 on DVM and PVM (Fig. 6d, e). These results confirm the MDR1 localization on DVM in the NSS[IP] parasites and on PPM in the NSR[NT] and NSS[NT] parasites.

### Variants of UBP1 display reduced Fluo-4 and MFQ accumulation in the parasite cytoplasm
*Plasmodium* MDR1 proteins have been implicated to transport drugs from the parasite's cytosol into the DV to alter parasite susceptibility to antimalarial drugs[7,36]. To investigate the substrate transport directions of the MDR1 in parasites carrying UBP1 IP and NT haplotypes, we engineered two parasite clones, NSS[IP/*mdr1*::mCherry] and NSS[NT/*mdr1*::mCherry] that had the N-terminus of the MDR1 fused with mCherry protein (Supplementary Fig. 6d-r, 6d-s). As expected, mCherry fluorescence was detected specifically at DVM and PPM in NSS[IP] and NSS[NT], respectively (Fig. 7a). Next, we investigated the accumulation of a known PfMDR1 substrate Fluo-4[9,37] in the NSS[IP/*mdr1*::mCherry] and NSS[NT/*mdr1*::mCherry] parasites. Whereas NSS[IP] principally displayed a uniform Fluo-4 accumulation in the parasite cytoplasm including the DV, in the NSS[NT] parasites, Fluo-4 mostly accumulated in the iRBC cytosol (~80% total signal), and pre-treatment with a P-gp inhibitor tariquidar (TQ)[9] significantly increased the amount of Fluo-4 in the parasite cytosol with concurrent reduction in the iRBC cytosol (Fig. 7a, b). For the NSS[IP] parasites, the majority of Fluo-4 signal was present in the parasite cytosol, and pre-TQ treatment decreased Fluo-4 in the parasite cytosol with the concurrent increase in the iRBC cytosol (Fig. 7a, c). As expected, the percentage of Fluo-4 in the DV was significantly reduced with TQ treatment (Fig. 7c). Additionally, we treated iRBCs with ACK lysing buffer to break up RBC membrane and then incubated the parasites with MFQ for 10 min. MFQ accumulation within parasite cells was measured using HPLC and LC-MS. Significantly lower MFQ was detected in the NSS[NT] parasites than in the NSS[IP] parasites (Fig. 7d). These results suggest that mis-located MDR1 on the plasma membrane

can mediate efflux of MFQ from parasite cytoplasm to iRBC cytosol in the NSS[NT] parasites.

### Inhibition of MDR1 reverses multiple drug resistances caused by UBP1 NT substitutions
In vivo, drug survival assays showed that NSS[IP] or NSS[NT] infected mice treated with TQ had 1.2- to 1.35-fold lower parasitemia (significant) than those treated with vehicle (Fig. 7e–j). Treatment of the NSS[IP] parasites with MFQ (20 mg/kg), LUM (10 mg/kg), and PPQ (30 mg/kg) with or without 12 mg/kg TQ completely killed the parasites (NSS[IP] is sensitive to the drugs, as expected) (Fig. 7e–g). For the NT mutant parasites, the same amounts of the drugs could significantly reduce parasitemia but were unable to clear the parasites (Fig. 7h–j). Co-administration of TQ with MFQ, LUM, or PPQ significantly reduced parasitemia (5.1- to 15.9-fold reduction) further to ~2% or lower (Fig. 7h–j). These results provide evidence that the P-gp inhibitor TQ can reverse UBP1[NT]-mediated multidrug resistances via inhibition of drug efflux out of parasite cytoplasm by MDR1 in vivo. Combinations of MDR1 inhibitors with antimalarial drugs transported by MDR1 could provide an avenue for developing new therapies.

## Discussion
The current study reveals a drug-resistant mechanism mediated by *Plasmodium* UBP1 and MDR1 that functions in two different biological pathways (Fig. 8). Two novel amino acid substitutions (I1560N and P2874T) in the *P. yoelii* UBP1 were shown to change MDR1 protein ubiquitination and localization and, therefore, enhance the parasite's ability to efflux compounds out of its cytoplasm and survive drug killing. Our study solves a long-standing puzzle on how mutations in a de-ubiquitinase confer drug resistance and opens a new field of study on drug resistance in malaria and other diseases. Indeed, screening for drugs to regulate protein ubiquitination and degradation as well as studying the drug-resistant mechanisms mediated by molecules in protein degradation pathways have received great attention in the therapeutic fields of cancer and infectious diseases in recent years[38–42]. Protein ubiquitination and proteasome-mediated protein degradation pathways were shown to play an important role in parasite drug responses[43], and ART treatment led to the accumulation of ubiquitinated proteins and parasite growth retardation[44]. *P. falciparum* responses to QN and quinidine (QD) were genetically linked to a HECT ubiquitin ligase (PF3D7_0704600) using progenies from a genetic cross[45]. More recently, knockdown protein expression of a *P. falciparum* Ring Finger Ubiquitin Ligase (PF3D7_1004300) was shown to reduce PfMDR1 protein levels and render the parasites more sensitive to DHA, PPQ, MFQ, and amodiaquine (AMQ)[46]. More research on protein ubiquitination and its effects on drug resistance, either directly on drug targets or indirectly on drug transporters, are required.

Two mutations (V2697F and V2728F) in the UCH domain of *P. chabaudi* UBP1 were shown to confer ART tolerance and CQ resistance[12–14], and the roles of the mutations in drug resistances have been confirmed experimentally[15,16]. Through genetic mapping for the unknown gene(s) conferring resistance to MFQ and allelic exchange experiments, here we found that simultaneously IP → NT substitutions in two key functional domains (CCR and UCH) of the *P. yoelii* UBP1 are required for high-level resistances to MFQ, LUM, and PPQ, but not CQ, QN, or DHA. These observations support the role of *Plasmodium* UBP1 in drug resistance; however, different mutations in the gene may lead to resistance to different drugs. For example, the V2697F and V2728F substitutions of *P. chabaudi* were linked to resistance to CQ and ART, but the I1560N and P2874T substitutions in *P. yoelii* did not significantly change parasite responses to CQ and DHA. As we report here, the IP → NT substitutions can affect the ubiquitination of many proteins including MDR1. Similarly, the V2697F and V2728F substitutions may also regulate the ubiquitination, trafficking, and function of other proteins such as CRT, K13, plasmepsins, or the amino acid

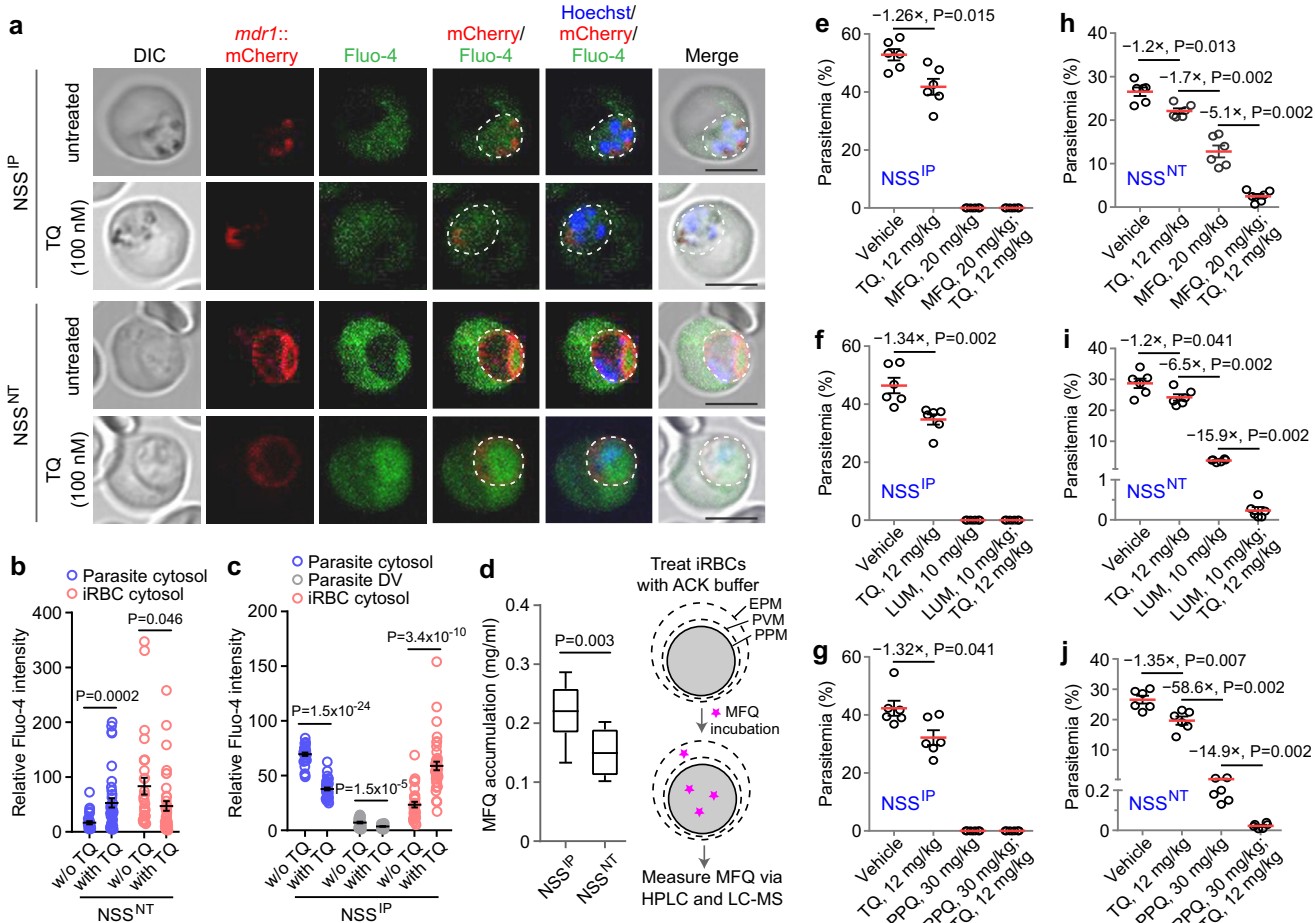

**Fig. 7 | Fluo-4 fluorescence accumulation and inhibition of MDR1 transport in NSS[IP] and NSS[NT] parasites. a** Representative images of *mdr1*::mCherry trophozoites after loading with Fluo-4 AM in the absence or presence of P-gp inhibitor tariquidar (TQ). Blue, Hoechst 33342. Scale bars, 5 μm. **b** Relative Fluo-4 intensity within infected RBC (iRBC) cytosol or parasite cytosol in NSS[NT] parasite with or without tariquidar (TQ) treatment. Mean ± SEM from 30 and 39 cells in the NSS[NT] (without TQ) and the NSS[NT] (with TQ) groups, respectively; two-tailed *t* test. **c** Relative Fluo-4 intensity within iRBC cytosol, parasite digestive vacuole (DV), or parasite cytosol in NSS[IP] parasite with or without TQ treatment. Mean ± SEM from 32 and 39 cells in the NSS[IP] (without TQ) and the NSS[IP] (with TQ) groups, respectively; two-tailed *t* test. **d** Comparison of mefloquine (MFQ) accumulation within isolated parasite cells after lysed with ACK buffer between NSS[IP] and NSS[NT] (*n* = 10 from three representative experiments; Mann-Whitney two-tailed *U*-test). In boxplots, the central line is the median; the top and bottom hinges correspond to the first and third quartiles, respectively; whiskers extend from the minimum to maximum values. Flow diagram (right panel) showing the MFQ accumulation assay performed on iRBCs. Cells were sequentially treated with ACK buffer to permeabilize the erythrocyte plasma membrane (EPM), incubated in the presence of MFQ (5 μM), and then subjected to HPLC and LC-MS analyses. PVM, parasitophorous vacuole membrane; PPM, parasite plasma membrane. In vivo efficacy tests in mice infected with NSS[IP] after administration of MFQ (**e**), lumefantrine (LUM) (**f**), and piperaquine (PPQ) (**g**) with or without TQ treatment. In vivo efficacy tests in mice infected with NSS[NT] after administration of MFQ (**h**), LUM (**i**), and PPQ (**j**) with or without TQ treatment. Mean ± SEM from 6 mice in each group in (**e**–**j**); Mann-Whitney two-tailed *U*-test, P values as indicated.

transporter AAT1[47,48] to influence parasite drug responses. There are approximately 30 predicted DUBs in the *Plasmodium* genome[49,50]. Although UBP1 is shown to deubiquitinate many parasite proteins, other DUBs may also affect parasite protein ubiquitination and responses to drugs. Additionally, mutations in PfCRT and copy number variation of plasmepsins 2/3 have also been linked to PPQ resistance[51,52]. The relationships of CRT, plasmepsins, MDR1, and UBP1 (or other DUBs) and their roles in drug resistance required further investigations.

Using allelic exchanged parasites, we demonstrated that the global ubiquitination was greatly increased in the UBP1 NT haplotype either with NSS or NSR background. In particular, 42 ubiquitination sites with nine 'up-regulated' and three 'down-regulated' were observed in the MDR1 protein (NSR/NSS). However, the mechanism of how the changed ubiquitination level of MDR1, or which specific ubiquitination sites, can alter MDR1 protein localization remains to be elucidated. Ubiquitination of PfMDR1 was also observed in *P. falciparum*[53,54], suggesting that a similar drug resistance mechanism

could emerge in *P. falciparum*. PfUBP1 variants were associated with ART-delayed parasite clearance in *P. falciparum* isolates from Kenya, Uganda, and Thailand, suggesting a role in responses to ART possibly by regulating other drug transporters/targets[17–19,55]. However, whether the UBP1 IP → NT substitutions can change MDR1 protein trafficking in human malaria parasites need to be determined in future studies.

Our results show that UBP1 functions as a de-ubiquitinase in vivo and that the IP → NT mutations greatly affect its de-ubiquitinating activity. The NT haplotype alters the intracellular expression pattern of UBP1, from focused punctate dots to diffused distribution within parasite cytosol. Given the high percentage of low complexity regions in the UBP1 and its puncta organization in the NSS parasite, it is possible that UBP1 can undergo self-association and drive phase separation through hydrophobic interaction. Indeed, treatment of iRBCs with 1,6-HD is known to dissolve biomolecular condensates[28,29] resulting in the alteration of UBP1 puncta to diffused pattern. The diffused distribution of the UBP1 NT variant suggests a reduced capacity for its self-association, leading to UBP1 protein misfolding and dysfunction.

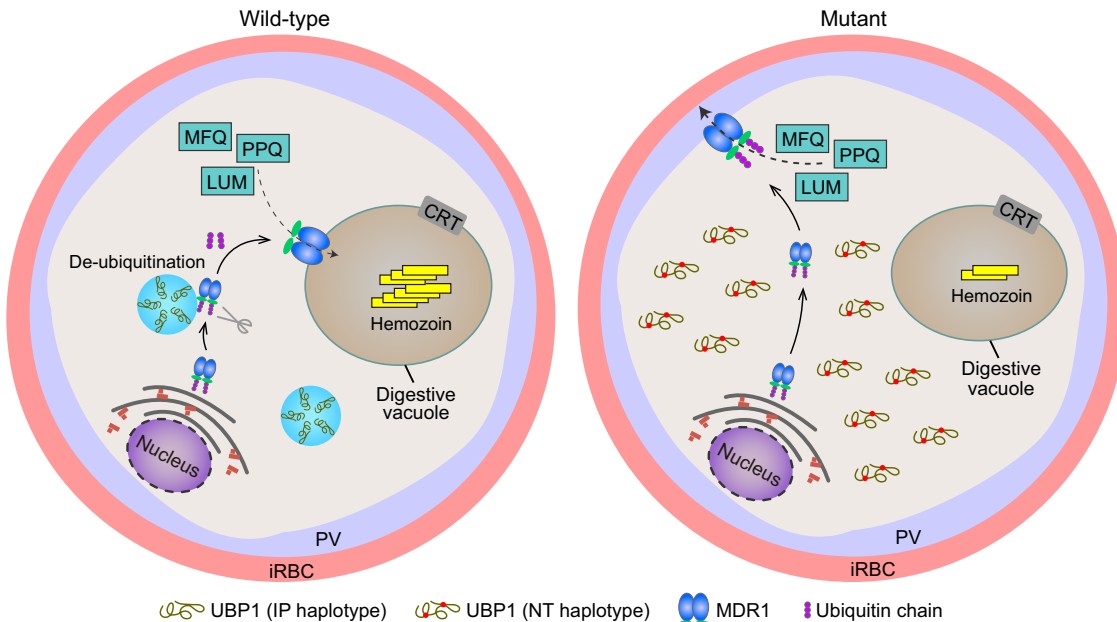

**Fig. 8 | Proposed mechanism of *Plasmodium* UBP1 and MDR1 mediated drug resistances.** In the wild-type parasite (IP haplotype), UBP1 in aggregated puncta functions to de-ubiquitinate parasite proteins including MDR1. Properly de-ubiquitinated MDR1 traffics to the digestive vacuole (DV) membrane where it may transport drugs such as mefloquine (MFQ), lumefantrine (LUM), and piperaquine (PPQ) into the DV. Accumulation of drugs in the DV may still be toxic to the parasites. In the mutant parasites (NT haplotype), UBP1 aggregate formation and de-ubiquitination activity is impaired due to the substitution of hydrophobic IP with hydrophilic NT amino acids, leading to increased MDR1 ubiquitination and altered MDR1 trafficking to the plasma membrane. The localization of MDR1 on the parasite plasma membrane allows the transport of drugs out of its cytoplasm resulting in drug resistance. iRBC, infected red blood cell; PV, parasitophorous vacuole.

Again, the relationships of specific amino acid substitutions, structural modification, protein trafficking, and function of UBP1 require more investigations.

UBP1 is one of the K13 interaction candidates (KICs) and has been linked to the endocytic uptake of hemoglobin from host cell cytosol, and a reduction in hemoglobin endocytosis results in ART resistance[20,56]. Interestingly, KIC1, KIC5, KIC6, MCA2, and K13 were among the top hit proteins with up-regulated ubiquitinated sites in NSR parasites compared with those of NSS parasites, which may influence parasite nutrient acquisition and/or parasite fitness. Our in vivo drug assay showed no effect of UBP1 IP→NT substitutions on parasite growth after ART treatment (Fig. 2c), which does not conflict with the 'ART resistant' phenotype based on ring survival assay (RSA) or delayed parasite clearance in vivo. The RSA is a phenotype only for a brief window (~6 h) of intra-erythrocytic development, whereas in our drug assays, the parasites were treated 4 doses over several days. The effects of the UBP1 mutations on the functions of these proteins also require further investigations. Nonetheless, our study reveals a previously unknown drug resistance mechanism involving two proteins functioning different pathways, providing a conceptual direction for future research on drug resistance in malaria and other diseases. The discoveries of consequential involvement of *Plasmodium* UBP1 and MDR1 in drug resistance may provide novel avenues for developing new drug combination therapies.

## Methods

### Parasites and infection of mice

Detailed information for key materials and reagents, including parasite lines, antibodies and chemicals, used in this study is provided in Supplementary Data 4. *P. yoelii* strains NSM and BY265 have been described previously[23]. NSM is an uncloned parasite that exhibits low-level MFQ resistance that was selected from *P. yoelii* NS, a parasite line that emerged from an isolate of *P. berghei* from Katanga, Belgian Congo, in the early 1970s[57,58]. NSR line was derived and cloned from

NSM after further MFQ (40 mg/kg) selection in our laboratory. NSS line was derived and cloned from the NSM after one passage through *Anopheles stephensi* mosquitoes. Interestingly, the NSS parasite became sensitive to MFQ after passing back to naïve mice from mosquitoes. The relationships and phenotypes of these parasites are summarized in Supplementary Fig. 1. Balb/c inbred mice and ICR outbred mice (female, 6–8 weeks old) used to maintain parasites, cloning, evaluate parasite growth and perform in vivo drug susceptibility assays, were purchased from Shanghai Laboratory Animal Center CAS (SLACCAS) or Xiamen University Laboratory Animal Center. The experimental procedures of mouse infection were performed according to protocols approved by the Animal Ethics Committees at Xiamen University (XMULAC20190050).

### *P. yoelii* genetic cross and linkage group selection

A colony of *Anopheles stephensi* mosquitoes (Hor strain) was used in the genetic cross experiments. All mosquitoes were raised at 24 °C and 75% humidity under a 12:12 light-dark illumination cycle and fad with 5% sucrose solution. The experimental procedures for producing genetic crosses in rodent malaria parasites have been described in detail previously[23,59]. Briefly, outbred ICR mice were co-infected with *P. yoelii* strains NSR and BY265 in a desired parental ratio according to the relative ability of the individual strain or line to produce gametocytes and oocysts[60]. Because BY265 produced ~7 times more gametocytes and ~10 times more oocysts than NSR (Supplementary Fig. 1f, g), a 5:1 to 10:1 (NSR/BY265) ratio in the number of inoculum parasites was used in the cross experiments. Female mosquitoes were fed on the donor mouse 4 days post-infection. Sporozoites were collected from the salivary gland of mosquitoes 16–18 days after the feeding and were injected intra-peritoneally (*ip*) into mice that were examined daily for the presence of blood-stage parasites. Infected mice were randomly divided into two groups, one subjected to MFQ (20 mg/kg) treatment for 5–8 days and one to no treatment. Uncloned progeny of the cross

from infected mouse blood was collected for DNA extraction and parasite preservation in a −80 °C refrigerator or liquid nitrogen. Genomic DNA from iRBCs was isolated using a phenol/chloroform method. DNA samples were typed with 190 microsatellites as described[23] using primers in Supplementary Data 1.

## Illumina HiSeq sequencing and SNP detection

Infected mouse blood samples were collected in 0.15% sodium citrate/PBS buffer and were passed through two NWF filters (Zhixin Bio, Bengbu, China) to remove host white blood cells. Parasite genomic DNA was isolated from the iRBCs pellet after the lysis with 0.05% saponin. A total of 2 μg DNA per sample was used for library preparation using TruSeq Library Construction Kit. Raw sequence data were processed to remove unusable reads using Illumina pipeline CASAVA v1.8.2. Sequence reads were aligned to the *P. yoelii* 17X reference genome (https://plasmodb.org/) using Burrows-Wheeler Aligner. SNP sets were called using Samtools and were filtered with mapping quality ≥20 & depth of the variate position ≥15.

## In vivo drug assay

The in vivo drug assays were performed according to Peter's 4-day drug test[57,58]. Briefly, mice were either injected (*ip*) with $10^5$ iRBCs and received drug treatment on Day 4 post-infection or injected intravenously (*iv*) with $10^6$ iRBCs and received drug treatment on Day 0 post-infection. Drugs (100 μl) were injected subcutaneously (*sc*) at the designated concentration for four consecutive days. Thin blood smears were prepared daily and stained with Giemsa stain. iRBCs were counted to calculate parasitemia (% of iRBCs). Mefloquine hydrochloride was dissolved in 70% ethanol; piperaquine phosphate in distilled water; chloroquine diphosphate, lumefantrine, dihydroartemisinin, and quinine hydrochloride dehydrate in DMSO and stored at −20 °C until use.

## Plasmid construction and CRISPR/Cas9 mediated gene editing

The methods used in allelic exchanges, gene tagging, and gene disruption were essentially as described[61,62]. To construct plasmids for *ubp1* allelic replacement at amino acid codon 1560 and 2874 between *P. yoelii* lines NSS and NSR, four DNA fragments from two regions (R1, from 4393 to 5036; and R2, from 8368 to 9034) were amplified and inserted into the NcoI/XhoI sites of the pYCm vector individually. Three silent nucleotide substitutions were introduced into the single guide RNA (sgRNA) binding sites in the donor templates using synthetic oligonucleotides and PCR amplification to yield the editing plasmids. To construct plasmids for gene tagging or tag insertion, a fragment (450–800 bp) from the C- or N-terminal of coding regions as left or right homologous arm and a fragment (450–800 bp) from 3′UTR or 5′UTR regions as right or left homologous arm were amplified, respectively. DNA sequence encoding 6HA, 4Myc, mCherry, GFP, APEX2, or 3HA was inserted between the left and right arms in the frame and cloned into the pYCm vector. To construct the plasmid to delete the coding region of the *P. yoelii ubp1* gene, 5′- and 3′- genomic fragments (400–600 bp) were amplified as left and right arms and cloned them into restriction sites of HindIII/NcoI and XhoI/EcoRI in the pYCm vector, respectively. Synthesized sgRNAs were annealed and ligated into the restriction site of BsmBI in the vector.

Electroporation of plasmids using the program T-016 in the Amaxa Nucleofector device, pyrimethamine selection of transformed parasites, negative selection using 5-fluorouracil to remove episomal plasmid for sequential modification, and parasite cloning were carried out as described[61]. Integration events were detected using PCR amplification and confirmed by DNA sequencing. Parasite cloning was followed if the integrated DNA sequences were detected. The sequences of the oligonucleotides and PCR primers for gene editing experiments are listed in Supplementary Data 5.

## Immunofluorescence assay (IFA)

iRBCs were fixed with 4% paraformaldehyde for 15 min and transferred onto a poly-L-Lysine pretreated coverslip. The fixed cells were permeabilized with 0.1% Triton X-100 in PBS for 10 min, blocked in 5% BSA solution for 1 h at room temperature, and subsequently incubated with the primary antibodies at 4 °C overnight. The samples were then incubated with fluorescently conjugated secondary antibodies for 1 h, and the nuclei of parasites were stained with Hoechst 33342 in the dark. Coverslips were mounted on glass slides in 90% glycerol solution and sealed with nail polish. Images were taken using identical settings on a Zeiss LSM 780 confocal microscope.

## Transmission electron microscopy imaging

APEX-based transmission electron microscopy (TEM) procedures were performed as described previously[63,64] with minor modifications. Briefly, iRBCs were harvested from mice that were infected with parasites expressing APEX2-fused UBP1, washed in 0.1 M PBS (0.081 M $Na_2HPO_4$, 0.019 M $NaH_2PO_4$, pH 7.4), and fixed with 2.5% glutaraldehyde solution at 4 °C overnight. The cells were then washed three times with 0.1 M PBS and embedded in an agarose sleeve. Blocks were immersed in 20 mM glycine at 4 °C for 10 min and then washed four times with 0.1 M PBS. Blocks were further incubated in DAB solution (0.5 mg/ml 3,3′-diaminobenzidine and 10 mM $H_2O_2$ in 0.1 M PB) for 1 h, followed by washing four times with 0.1 M PBS. The cells were stained with 2% $OsO_4$ solution in 0.1 M PBS, and 2% uranyl acetate solution in $ddH_2O$, and embedded in resin before slicing into thin sections (70–80 nm in thickness). Images were captured using transmission electron microscopy (Hitachi, HT-7800).

## Fluo-4 loading and live cell imaging

iRBCs were collected from mice that were infected with parasites expressing mCherry-tagged MDR1. After the removal of blood plasma by centrifugation at $400 \times g$ for 5 min, cells were washed with Ringer's solution (122.5 mM NaCl, 5.4 mM KCl, 1.2 mM $CaCl_2$, 0.8 mM $MgCl_2$, 11 mM G-glucose, 10 mM HEPES, 1 mM $NaH_2PO_4$, PH 7.4) three times, loaded with 5 μM Fluo-4 AM for 1 h at 37 °C, and were transferred to a 20 mm culture dish (#801001, NEST) as described previously[9,37]. For inhibition of Fluo-4 transport, cells were pre-incubated with 100 nM tariquidar for 10 min before adding the Fluo-4 AM probe. Samples were washed with Ringer's solution and placed at 37 °C for 20 min. After staining nuclei with Hoechst 33342, the cells were imaged using Zeiss LSM 780 confocal microscope.

## 1,6-HD treatment and puncta analysis

iRBCs (parasites expressing GFP-tagged UBP1) were treated with 10% of 1,6-HD in PBS for different time periods. The cells were washed with PBS 3X after the removal of 1,6-HD. After staining nuclei with Hoechst 33342, the cells were processed for imaging. The puncta number inside each cell was calculated after filtering with the same threshold by the Analyze Particles module of ImageJ software. The UBP1 signal area per cell was measured, and the number was transferred to GraphPad Prism for statistical analysis. More than 30 cells were used for quantification analysis from each blood stage of parasites.

## Hemozoin quantification

Thin blood smears were prepared from infected mouse tail blood and stained with Hoechst 33342 for 20 min. Hemozoin inside iRBC was quantified by measuring the relative intensity (RI) of hemozoin crystals using a reflection contrast polarized light microscopy (Olympus IX83). The RI of hemozoin crystals was analyzed using ImageJ software. At least 40 schizonts (>5 nuclei) were measured for each parasite.

## MFQ accumulation assay in isolated parasites

Infected mouse blood was collected in a heparin tube; the red blood cell membrane was lysed with Ammonium-Chloride-Potassium (ACK)

lysing buffer (150 mM $NH_4Cl$, 10 mM $KHCO_3$, 0.1 mM $Na_2EDTA$). After the removal of supernatant fluid by centrifugation at $850 \times g$ for 2 min, parasite pellets were washed three times with PBS. Approximately $2.5 \times 10^8$ parasite cells were incubated in 2 ml RPMI 1640 medium containing 10% fetal bovine serum and 5 μM MFQ at 37 °C for 10 min. The parasite cells were harvested from the culture medium by centrifugation at $15,000 \times g$ for 15 min and transferred into a 1.5 ml Eppendorf tube containing 980 μl extraction buffer (Methanol:Acetonitrile:$ddH_2O$ = 2:2:1). The suspension was ultra-sonicated 5 cycles each for 3 s and then incubated at −20 °C for 1 h. The MFQ concentration was measured using an Agilent 1260 Infinity HPLC system with a reversed-phase column (HD-C18, 5 μm, ZhongPu), and mass spectra were collected using a Waters-515 LC-MS system equipped with C18 analytical column (4.6 × 50 mm, 5 μm).

### In vivo drug tests of combinations with tariquidar
Mice were injected (*iv*) with $2 \times 10^5$ iRBCs and treated with various regimens on Day 0 post-infection for four consecutive days. Infected mice were injected (*sc*) with mefloquine (20 mg/kg), lumefantrine (10 mg/kg), or piperaquine (30 mg/kg). Tariquidar (12 mg/kg) or vehicle (5% glucose) was administered (*ip*) 2 h before the antimalarial drug treatment. Thin blood smears were made, and parasitemias were monitored daily. Tariquidar was dissolved in 5% (w/v) glucose solution and stored at −20 °C until use.

### Protein extraction and Western blotting
Infected mouse blood was collected and lysed with 0.1% saponin in PBS on ice, and parasite pellets were washed three times with PBS. Proteins were extracted using lysis buffer LysC (20 mM Tris-HCl, 50 mM NaCl, 1 mM DTT, 1% Triton X-100, pH 8.0) with protease inhibitor cocktail and PMSF. After incubation on ice for 30 min, the samples were centrifuged at $15,000 \times g$ for 15 min at 4 °C. Proteins were resolved on SDS-PAGE gels with appropriate concentration according to the theoretical molecular weight of target proteins and transferred to the PVDF membrane. The membranes were blocked with 5% skim milk in TBST buffer at room temperature for 1 h and probed with primary antibody at 4 °C overnight. Membranes were then incubated with horseradish peroxidase (HRP)-conjugated secondary antibody for 1 h at room temperature. The Western blotting was done using enhanced chemiluminescent (ECL) reagents, and images were processed using the Clinx Science Instruments chemiluminescence system (ChemiScope 6200).

### Immunoprecipitation
Parasite pellets were prepared as described above and lysed with 1 ml LysC plus protease inhibitor cocktail and PMSF on ice for 30 min before centrifugation at $15,000 \times g$ for 15 min at 4 °C. One microliter of mouse anti-Myc antibody (cat#2276, CST) or rabbit anti-HA antibody (cat#3724, CST) was added to the supernatant, and the solution was incubated on a vertical mixer for 16 h at 4 °C. Thirty μl of protein A/G beads (HY-K0202, MedChemExpress) that pre-balanced with LysC was added, incubated for 5 h at 4 °C, and centrifuged at $400 \times g$ for 2 min at 4 °C to remove the supernatant. The beads were then washed three times with LysC at 4 °C, and bound proteins were eluted using 80 μl 2.5 × SDS loading buffer.

### Mass spectrometry and Ub-modified proteome (ubiquitinome) analysis
Protein samples were prepared from mixed blood stages (~3% ring, ~35% trophozoite, and ~57% schizont) of parasites after harvested by centrifugation using 60% Nycodenz/PBS solution, iRBCs were lysed with 0.1% saponin on ice. Parasite pellets were subjected to sonication in lysis buffer (8 M urea, 1% protease inhibitor cocktail), and supernatants were collected after centrifugation at $15,000 \times g$ for 10 min at 4 °C. The samples were digested at a 1:50 enzyme-to-protein ratio (w/w) using trypsin overnight at 37 °C. To enrich Ub-modified peptides, tryptic peptides dissolved in NETN buffer (100 mM NaCl, 1 mM EDTA, 50 mM Tris-HCl, 0.5% NP-40, pH 8.0) were incubated with pre-washed antibody-conjugated beads (PTM-1104, PTM Bio) overnight at 4 °C with gentle shaking. The beads were successively washed with NETN buffer four times and with $ddH_2O$ twice. Peptides were then eluted from the beads using 0.1% trifluoroacetic acid. The eluted peptides were subjected to a nano-electrospray ion source followed by tandem mass spectrometry (MS/MS) in Q Exactive™ Plus (Thermo) coupled online to the UPLC. The LC-MS/MS analysis in this study was supported by Jingjie PTM BioLabs.

### Statistical analysis
Sample size or independent biological replicates were listed in the text and figure legends. Statistical analysis was tested using a two-tailed *t* test or Mann-Whitney *U*-test in Excel or GraphPad Prism. *P* values less than 0.05 were considered significant. *n* represents the sample volume in each group or the number of biological replicates.

### Reporting summary
Further information on research design is available in the Nature Portfolio Reporting Summary linked to this article.

## Data availability
The Illumina sequencing data generated in this study have been deposited in the Genome Sequence Archive in National Genomics Data Center, China National Center for Bioinformation under the accession number: CRA013922. Mass spectrometry proteomics data have been deposited to the ProteomeXchange Consortium via the PRIDE partner repository with the dataset identifier: PXD047782 and PXD047811. Source data are provided with this paper.

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

## Acknowledgements

This work was supported by the National Natural Science Foundation of China (82072302), the Natural Science Foundation of Fujian Province of China (2022J01028), and the National Parasitic Resources Center/the Ministry of Science and Technology Fund (NPRC–2019-194-30) to J.L., and partially by the Division of Intramural Research, National Institute of Allergy and Infectious Diseases, National Institutes of Health (to X-z.S.). The authors thank Dr. Bo Wang for helpful discussion, Xinyu Wu for experimental assistance, and Yolanda L. Jones, NIH Library Editing Services, for editing assistance.

## Author contributions

R.X. and L.L. performed most of the experiments and analyzed data; Z.J. contributed to drug assay and sample processing; R.L., Y.G., Y.Z. and X.W. provided assistance or technical support in some experiments; X.S. and Y.W. performed HPLC and LC-MS experiments under the supervision of X.D.; L.Y. provided assistance in the TEM imaging experiment; S.L. and J.Y. analyzed data and coordinated the work; J.L. supervised the project; X-z.S. and J.L. conceived the project, designed experiments, analyzed data, and wrote the manuscript.

## Competing interests

The authors declare no competing interests.
