## [Peer Review File · Nature Communications]

Deaggregation of mutant Plasmodium yoelii de-ubiquitinase
UBP1 alters MDR1 localization to confer multidrug resistanceREVIEWER COMMENTS

Reviewer #1 (Remarks to the Author):

Xu and colleagues have presented a manuscript describing results from an important study of rodent malaria parasites selected by in vivo drug exposure, designed to generate reduced parasite susceptibility to the antimalarial mefloquine. These parasites are shown to be also significantly less susceptible to lumefantrine and piperazine. This is a very important result.

Evidence is presented that this phenotype is mediated by newly described sequence variants at the *ubp1* locus of *P. yoelii*, and that the mechanism involves disaggregation of punctate UBP1 complexes in the cytoplasm and aberrant ubiquitination of the crucial digestive vacuole transporter PFMDR1. This appears to result in the inappropriate trafficking of *pfmdr1* to the parasite plasma membrane, where it is speculated it is able to efflux antimalarial drugs.

This is a rich and complex study which is potentially a major advance in our understanding of the role of UBP1 in the cell, and how sequence variants of this protein impact drug susceptibility in many different ways. The paper makes a significant contribution to our knowledge, but I do have some reservations and suggestions for improvements in the text and presentation.

A general criticism is that the authors do not consider the importance of asexual parasite stage. Evidence from *P. falciparum* is that UBP1 variants are modulators of endocytosis of haemoglobin during the first 6h post-invasion, and that this influences artemisinin susceptibility (as set out in line 389-395 of the Discussion), but only for this brief window of intra-erythrocytic development. In contrast, the PFMDR1 plasma membrane mislocalisation model for mef/lum/pip resistance set out by the authors is likely to be completely unrelated. Can the authors provide some consideration of the life cycle stage(s) at which they predict drug efflux by plasma membrane PFMDR1 will be most important for the resistance phenotypes they describe in *P. yoelii*?

A second criticism is that the penicillin-binding protein domain identified is not fully defined or explained. This is the first structural element of the very large UBP1 protein to be described, outside of the canonical cysteine peptidase/de-ubiquitinase domain at the carboxy terminus of the protein sequence. What are the distinguishing PBP sequence motifs? What are the possible functions of such a domain (described in prokaryotes mainly) in a protozoan cell?

Other Comments

ABSTRACT

Line 21 - the phrase "are required" here implies that the NT haplotype is the only way to achieve this. It would be more precise to write: "together generate high-level resistance ..." or something similar.

Line 22 - These statements are a little too strong, and I suggest moderation. For example:

"Evidence is presented that these mutations together impair UBP1 cytoplasmic aggregation ... etc"

INTRODUCTION

Lines 55-60 - This is probably best written in the present or perfect tense. I suggest: "In this study, two new mutations in *Plasmodium yoelii* UBP1 (I1560N and P2874T) are identified and linked ... Evidence is presented that *P. yoelii* MDR1 is a substrate of UBP1, and that these mutations perturb UBP1 cytoplasmic aggregation ..." and so on.

RESULTS

Line 94 - In my view, it is important to point out here that in *Py* (and all rodent malaras I think) chromosome 2 is syntenic with chromosome 1 of *P. falciparum*, due to a substantial deletion on one arm of the *Py* chromosome syntenic with *Pf* chromosome 2, such that by PFG this chromosome ran as the smallest, and hence was designated #1. Thus the *ubp1* locus would have been predicted to be on *Py* chromosome 2, and is probably surrounded by extended synteny with *Pf* chromosome 1. (I checked quickly on Plasmo DB and this seems to be the case).

Line 106 -Please provide codon numbers here in the text.

Line 114 - some further info and evidence here (or Supplementary?) on the PBP would be useful (see my comment above).

Lines 126-142 - this series of constructs was very well conceived and provide an excellent panel

for all the investigations that follow. The experiments in Figure 2 are informative and the results clear.

Line 139 - Does NSS-NT have a fitness problem? Can this be tested in vitro (I guess in Pf)? It might suggest other changes in NSR parasites were important for mitigating fitness loss due to *ubp1* mutations. Do you have any evidence of this?

Line 145 - "mild increase" here is not informative. What were you measuring? What went up? EC50? or susceptibility? If latter, then EC50 went down. Please use precise language.

Lines 167-174 - these experiments are useful and the results reasonably compelling. However, as this is a general ubiquitination signal you are measuring, I am surprised that modifications to one DUB enzyme (UBP1) has a global effect on total cellular levels of this modification. Does this suggest that UBP1 is the major blood-stage DUB? How many DUB-encoding genes have been identified in the Plasmodium genome? Perhaps a comment in the Discussion is warranted.

Line 189 - Fig 4 provides good evidence to support the punctate distribution of UBP1, but I think you should state "in the cytoplasm of blood-stage trophozoites and schizonts" here to emphasise that two stages were evaluated. Can you provide here or in the Fig 4 legend the approximate age range (post-invasion, in hours) of these two stages in *P. yoelii*?

Line 207 - To what stage of development was the 1,6-HD applied? I have some concern that this compound has a GENERAL effect on biomolecular aggregates in the cell. You should qualify your findings by saying that you can't rule out off-target impacts with such a general cellular modulator as 1,6-HD, unless your controls address this and I missed it.

Line 216 - I approve of the cautious language in this sentence. Your work supports some important hypotheses, but these remain to be proven. The self-association idea is important, and I hope further work on this phenomenon in UBP1 is planned.

Line 237 - I suggest you replace the words "than those of" with "compared to the levels of these peptides in the" ...

Lines 253-256 - Can you explain the extra (smaller) K13 bands in the NSR-NT lane of fig 5f?

Line 256 - for the crt ubiquitination experiment, is it possible that crt ubiquitination occurs during a narrow time-window in the parasite? Is it possible that the parasite preparation used here did not encompass all of the intra-erythrocytic life-cycle?

Lines 267-269 - Although it is consistent across Fig 6a and 6b, it does not appear that the change in % MDR1 localisation between the two -NT parasite lines after MQ administration is significant. Did you apply a statistical test here? My instinct would be to moderate or remove this statement of the impact of MQ pressure on this localisation measure, unless you have statistical confirmation.

Lines 277-283 - Please put in present tense. These sentences are a little clumsy and need improving to provide better clarity. When you write "suggesting similar temporal expression of the two proteins" do you really mean to say "suggesting that adding a second tag did not alter temporal expression of UBP1"?

Line 289 - I suggest this subtitle should be "Variants of UBP1 display reduced Fluo-4 ..."

Line 308 - first word "were" should be "was"

Line 309-311 - Is it better to say "These results suggest that mis-located MDR1 on the plasma membrane can mediate efflux of MFQ from parasite cytoplasm ... "

Line 322 - I suggest replace "confirm that" with "provide evidence that"

DISCUSSION

Line 362 - delete "recently reported" as this *aat1* gene was first flagged in *P. chabaudi* by Hunt and colleagues in a publication in 2012. You can retain the Ngwa et al reference however - it is appropriate.

Line 371 - The statement is too strong. I suggest replacing "may also exist" with "could emerge"

FIGURE LEGENDS

The Figures are mostly excellent, but the legends lack sufficient information and precisions in some cases. Please review these carefully.

Line 616 - in westerns, please name the antibody specificity in each case. For 3e please explain what the y-axis units are - "ubiquitinated ratio" - I can see this is normalised to the level in the IP parasites, but please spell this out the first time it is used.

Lines 618 and 620 - you should name the technique (western blot I assume) each time if there is any ambiguity

Line 648 - please add the phrase ""from TEM images such as those in (h)" [NB it is not (g)]... this should come after the word "puncta".

Line 658 - Fix this, for example: "Proteins with the most regulated ubiquitination sites (≥ 3) in NSR compared to NSS parasites;" ...

Line 661 - specify the tag used for IP and the Ab for the pull-down (I guess it is ubiquitin and anti-ubiquitin but it still needs to be stated)

Line 663 - specify the Ab used for the immunoblotting

Colin Sutherland
LSHTM
London

Reviewer #2 (Remarks to the Author):

This is a comprehensive and exceedingly thoroughly-controlled study on the function of the Plasmodium UBP1 deubiquitinating enzyme.

UBP1 has been long associated with parasite drug resistance and point mutations within its sequence have become markers for artemisinin resistance in field isolates. To date, no substrates or biological function have been attributed to this enzyme beyond it being localised to the parasite cytosome. As such, this study will undoubtedly be of considerable interest to the wider malaria community and will be important in influencing further work into how drug resistance arises.

The authors identified two point mutations that spontaneously arose by cycling parasites under drug pressure. These mutations together conferred a fitness defect (interesting in itself considering other point mutant do not have this defect), an andocytic defect as measure by lower haemozoinb in the food vacuole, higher systemic ubiquitination and mislocalisation of the UBP1 protein.

Ubiquitinome analysis revealed MDR1 was differentially ubiquitinated along with a set of Kelch13 interacting proteins. The authors demonstrated with a set of elegant approaches and convincing experiments that in the absence of UBP1-mediated deubqiutination, MDR1 mislocalises from the FV to the PM where is serves as a drug efflux pump, thereby conferring drug resistance to the parasite.

I find the data convincing, well-controlled, and of great significance to malaria cell biology. As such, I only have minor comments and corrections to suggest.

Minor corrections/comments:

The penicillin binding domain is novel but no explanation is given as to what it might be doing. It would be helpful if the authors discussed it more thoroughly.

Some of the IFA figures are distorted- these need to be amended to publication (example figure 4F)

Figure 6a/b: the authors show that MDR1 is largely mislocalised to the PM in the presence of the UBP1 double mutant but that this effect is even more pronounced under drug pressure. What do they propose the mechanisms of this additional drug-induced effect is?

The data presented pertaining to the self-association of UBP1 being necessary for proper localisation was a bit over interpreted. Are there other examples of DUBs in other biological systems where oligomerisation is necessary for activity? Disucing such cases would strengthen the arguments.

If MDR1 is indeed functioning as a drug efflux pump at the PPM, and given its prior association with DHA and artemisinin resistance, it seems strange that DHA resistance is unaffected in this study. The authors touch upon this in the discussion by saying different point mutation in UBP1 may give rise to different drug resistance profiles. Would this suggest that MDR1 is not actually involved in DHA/ART resistance?

Reviewer #3 (Remarks to the Author):

The manuscript on mutant *ubp1* relocalizing *mdr1* in the mouse model is a tour de force with well crafted studies laid out in a logical manner with a thorough approach to nailing the evidence. The title should note genus and species. Many drug resistant mechanisms differ between *P. vivax* or *falciparum* and the murine *Plasmodium*. The last line of the abstract also needs to contain the words "in a mouse model" after substrate transport direction, in a mouse model, providing... While the authors noted in line 103 on page 5 no mutation or copy number variation in orthologues of *Pfcr1*, *pfmdr1*, *pfmdr2* and *pfk13* they left out *plasmepsin*. This copy number or mutation should be looked for in existing data as in *P. falciparum* they are often opposites in number of copies with piperquine resistance with increase CNV in *plasmepsin* having low CNV in *pfmdr1* and vice versa. The results section on page 7 line 150 to 153 need to explicitly note if resistance to both mefloquine and piperaquine tracked together or diverged amongst the mutants in NSS and NSR. This reviewer favors identical linear or log x axis for figure 2C as the DHA decreasing by 10 mg/kg is confused with mfq or ppq with x axis 40 or 60. I would add a table of mg/kg for IC50, IC90 and IC99 for isolates and drugs to clarify the figure 2 C phenotypes by drugs.

Plasmepsins can be brought into the discussion

Responses to Reviewers' comments:

REVIEWER COMMENTS

Reviewer #1 (Remarks to the Author):

Xu and colleagues have presented a manuscript describing results from an important study of rodent malaria parasites selected by in vivo drug exposure, designed to generate reduced parasite susceptibility to the antimalarial mefloquine. These parasites are shown to be also significantly less susceptible to lumefantrine and piperaquine. This is a very important result.

Evidence is presented that this phenotype is mediated by newly described sequence variants at the ubp1 locus of P. yoelii, and that the mechanism involves disaggregation of punctate UBPI complexes in the cytoplasm and aberrant ubiquitination of the crucial digestive vacuole transporter PFMDR1. This appears to result in the inappropriate trafficking of pfmdr1 to the parasite plasma membrane, where it is speculated it is able to efflux antimalarial drugs.

This is a rich and complex study which is potentially a major advance in our understanding of the role of UBPI in the cell, and how sequence variants of this protein impact drug susceptibility in many different ways. The paper makes a significant contribution to our knowledge, but I do have some reservations and suggestions for improvements in the text and presentation.

Thank you for the positive comments.

A general criticism is that the authors do not consider the importance of asexual parasite stage. Evidence from P. falciparum is that UBPI variants are modulators of endocytosis of haemoglobin during the first 6h post-invasion, and that this influences artemisinin susceptibility (as set out in line 389-395 of the Discussion), but only for this brief window of intra-erythrocytic development. In contrast, the PFMDR1 plasma membrane mislocalisation model for mef/lum/pip resistance set out by the authors is likely to be completely unrelated. Can the authors provide some consideration of the life cycle stage(s) at which they predict drug efflux by plasma membrane PFMDR1 will be most important for the resistance phenotypes they describe in P. yoelii?

Response: This is a good point. Indeed, the parasites were synchronized *in vitro* before performing a ring survival assay (RSA) for *P. falciparum*. For *P. yoelii*, it is difficult to synchronize the parasites *in vivo*. To address this issue, we have added: “Our *in vivo* drug assay showed no effect of the UPB1 IP→NT substitutions on parasite growth after ART treatment (Fig. 2c), which does not conflict with the ‘ART resistant’ phenotype based on ring survival assay (RSA) or delayed parasite clearance *in vivo*. The RSA is a phenotype only for a brief window (~6h) of intra-erythrocytic development, whereas in our *in vivo* drug assays, the parasites were treated 4 doses over several days” in the Discussion. Line 422-427.

A second criticism is that the penicillin-binding protein domain identified is not fully defined or explained. This is the first structural element of the very large UBPI protein to be described, outside of the canonical cysteine peptidase/de-ubiquitinase domain at the carboxy terminus of the protein sequence. What are the distinguishing PBP sequence motifs? What are the possible functions of such a domain (described in prokaryotes mainly) in a protozoan cell?

Response: Thank you for pointing this out. Using the Swiss-Model server, we previously identified a UBPI amino acid region harboring the I1560N mutation that has some homology with the motifs III-IV of the glycosyltransferase (GTase) domain in bacterial Penicillin-binding proteins (PBPs) (**Fig. S4a**). The PBPs are bifunctional proteins containing GTase activity and transpeptidase (TPase) activity for peptidoglycan synthesis in bacteria including *Escherichia coli* and *Aquifex aeolicus* (doi: 10.1016/j.jmb.2008.08.020). However, the UBPI region shared only ~30% similarity with the GTase domain of the bacterial PBPs. Additionally, the UBPI 'KKKIMDEILNTL' motif (motif IV of bacterial PBP1) is also homologous to an ATP-binding cassette sub-family G member 1-like isoform X1 of *Belonocnema kinseyi* (**Fig. S4b**) and a DNA repair protein of *Campylobacter concisus* (**Fig. S4c**). We, therefore, decided to replace the "PBP domain" with "conserved central region (CCR) domain" in the text due to the lack of further functional characterization of the domain.

We have modified the sentence to read: "The NT mutations were located within a conserved central region (CCR) and ubiquitin carboxyl-terminal hydrolase (UCH) domains (Supplementary Fig. 3), respectively. The CCR has some homology with the motifs III-IV of the glycosyltransferase (GTase) domain of bacterial penicillin-binding proteins (PBPs) (Supplementary Fig. 4a), as predicted using the Swiss-Model automated server (<https://swissmodel.expasy.org/>). The PBPs are bifunctional proteins containing GTase activity and transpeptidase (TPase) activity for peptidoglycan synthesis in bacteria including *Escherichia coli* and *Aquifex aeolicus*²⁵. However, the CCR region shared only ~30% similarity with the GTase domain of the bacterial PBPs. Additionally, the UBPI 'KKKIMDEILNTL' motif (motif IV of bacterial PBP1) is also homologous to an ATP-binding cassette sub-family G member 1-like isoform X1 of *Belonocnema kinseyi* (Supplementary Fig. 4b) and a DNA repair protein of *Campylobacter concisus* (Supplementary Fig. 4c). The function of the UBPI CCR is uncertain, and further investigation is required. To our knowledge, the IP→NT substitutions were identified for the first time, and the CCR domain has not been characterized in malaria parasites previously."

Line 115-130

a

PBP1b E. coli	271	QLVKNLFLSSERSYWRKANEAYMALIMDARYSKDRILELYMNEVYLGQSGDNEIRGFPLA	330
PBP1a E. coli	124	QLARNFFLSPERTLMRKIKEVFLAIRIEQLLTKDEILELYLNKIYLGYRAYGVGAAAQVY	183
PBP1a A. aeolicus	121	QLAKNLF LTRERTLERKIKEALLA IKIERTFDKKKIMELYLNQIYLGSGAYGVEAAAQVY	180
NSS	1517	KIVKNILLSLELTNDNKISKIRKILFYSPSEEEKKIMDEILNTLYMYPQLYVSCIISLFY	1576
NSR	1517	KIVKNILLSLELTNDNKISKIRKILFYSPSEEEKKIMDEILNTLYMYPQLYVSCIISLFY	1576
17XL	1434	KIVKNILLSLELTNDNKISKIRKILFYSPSEEEKKIMDEILNTLYMYPQLYVSCIISLFY	1493
BY265	1458	KIVKNILLSLELTNDNKISKIRKILFYSPSEEEKKIMDEILNTLYMYPQLYVSCIISLFY	1517
PbANKA	1421	KIVKNILLSLELTNDNKISKIRKILFYSPSEEEKKIMDEILNTLYMYPQLYVSCIICLFY	1480
PcAS	1378	KIVKNILLSLELTNDNKINKIRKILFYSSSEEEKKIMDEILNTLYIYPQLYVSCIICLFY	1437
Pf3D7	1744	KNVKNILLSLELSNEEKINEVRKILFYSSSDEKKYIMNEILNLIYIYPQLYVSCIISLFY	1803

Homology to glycosyltransferase domain of the bacterial Penicillin-Binding Proteins (PBPs)

b

Query	1	KKKIMDEILNTL	12
		K KIMDE+LN L	
Sbjct	132	KQKIMDEVLSL	143

ATP-binding cassette sub-family G member 1-like isoform X1 (*Belonocnema kinseyi*)

c

Query	2	KKIMDEILNT	11
		KKIMDEIL T	
Sbjct	133	KKIMDEILKT	142

DNA repair protein (*Campylobacter concisus*)

Supplementary Fig. 4. Sequence alignment to bacterial Penicillin-Binding Proteins (PBPs) and a conserved UBPI motif. **(a)**, Partial sequence homology of *Plasmodium* UBPI amino acid sequence with bacterial Penicillin-Binding Proteins (PBPs) was predicted using the Swiss-Model automated server (<https://swissmodel.expasy.org/>) and aligned using Jalview (<https://www.jalview.org/>). **(b)**, Sequence alignment and partial homology to an ATP-binding cassette using BLAST searches. **(c)**, Sequence alignment to a motif in DNA repair protein using BLAST searches.

Other Comments

ABSTRACT

Line 21 - the phrase "are required" here implies that the NT haplotype is the only way to achieve this. It would be more precise to write: "together generate high-level resistance ..." or something similar.

Response: We replaced "are required" with "can mediate".

Line 22 - These statements are a little too strong, and I suggest moderation. For example: "Evidence is presented that these mutations together impair UBPI cytoplasmic aggregation ... etc"

Response: We changed the sentence to read: "Mechanistically, the double mutations were shown to ..."

INTRODUCTION

Lines 55-60 - This is probably best written in the present or perfect tense. I suggest: "In this study, two new mutations in *Plasmodium yoelii* UBPI (I1560N and P2874T) are identified and linked ... Evidence is presented that *P. yoelii* MDR1 is a substrate of UBPI, and that these mutations perturb UBPI cytoplasmic aggregation ..." and so on.

Response: Done as suggested, thanks.

RESULTS

Line 94 - In my view, it is important to point out here that in *Py* (and all rodent malarial *I*

think) chromosome 2 is syntenic with chromosome 1 of P. falciparum, due to a substantial deletion on one arm of the Py chromosome syntenic with Pf chromosome 2, such that by PFG this chromosome ran as the smallest, and hence was designated #1. Thus the ubp1 locus would have been predicted to be on Py chromosome 2, and is probably surrounded by extended synteny with Pf chromosome 1. (I checked quickly on Plasmo DB and this seems to be the case).

Response: We have added: “...that is syntenic with chromosome 1 of *P. falciparum*²⁴”.

Line 106 -Please provide codon numbers here in the text.

Response: We added the codon numbers as suggested. “...resulting in Ile→Asn (I→N, codon #1560) and Pro→Thr (P→T, codon #2874) amino acid...” **Line 108.**

Line 114 - some further info and evidence here (or Supplementary?) on the PBP would be useful (see my comment above).

Response: Please see the responses to the ‘a second criticism’ above.

Lines 126-142 - this series of constructs was very well conceived and provided an excellent panel for all the investigations that follow. The experiments in Figure 2 are informative and the results clear.

Response: Thank you!

Line 139 - Does NSS-NT have a fitness problem? Can this be tested in vitro (I guess in Pf)? It might suggest other changes in NSR parasites were important for mitigating fitness loss due to ubp1 mutations. Do you have any evidence of this?

Response: Yes, the difficulty in cloning the NT double mutant suggests that the parasite has a fitness problem. When we clone parasites from a mixture, we tend to clone the fast growers. That was why we decided to select the parasites with mefloquine. Luckily, we were able to obtain the NT double mutant after mefloquine selection. Indeed, the NSR parasite grew slower than the NSS parasite without drug selection (Please see Supplementary Figure 1c). We have added: “This result is consistent with the finding that NSR parasite grew slower than the NSS parasite without drug selection (Supplementary Fig. 1c)” to support this claim. **Line 154-155.**

Line 145 - "mild increase" here is not informative. What were you measuring? What went up? EC50? or susceptibility? If latter, then EC50 went down. Please use precise language.

Response: We have replaced “mild increase” with “slight increase in IC₉₀”.

Lines 167-174 - these experiments are useful and the results reasonably compelling. However,

as this is a general ubiquitination signal you are measuring, I am surprised that modifications to one DUB enzyme (UBP1) have a global effect on the total cellular levels of this modification. Does this suggest that UBP1 is the major blood-stage DUB? How many DUB-encoding genes have been identified in the Plasmodium genome? Perhaps a comment in the Discussion is warranted.

Response: Indeed, UBP1 appears to affect the ubiquitination of many proteins and is a major DUB in the blood-stages. There are 29 predicted DUBs in the *P. falciparum* genome (doi: 10.1371/journal.pone.0002386; doi: 10.3389/fcimb.2022.985178). We have added: “There are approximately 30 predicted DUBs in the *Plasmodium* genome^{49,50}. Although UBP1 is shown to deubiquitinate many parasite proteins, other DUBs may also affect parasite protein ubiquitination and responses to drugs. Additionally, mutations in PfCRT and copy number variation of plasmepsins 2/3 have also been linked to PPQ resistance^{51,52}. The relationships of CRT, plasmepsins, MDR1, and UBP1 (or other DUBs) and their roles in drug resistance required further investigations”. **Line 386-392.**

Line 189 - Fig 4 provides good evidence to support the punctate distribution of UBP1, but I think you should state "in the cytoplasm of blood-stage trophozoites and schizonts" here to emphasize that two stages were evaluated. Can you provide here or in the Fig 4 legend the approximate age range (post-invasion, in hours) of these two stages in P. yoelii?

Response: Changed as suggested (**blood-stage trophozoites and schizonts**). Since the parasites were not synchronized, it is difficult to give an estimate of time.

Line 207 - To what stage of development was the 1,6-HD applied? I have some concerns that this compound has a GENERAL effect on biomolecular aggregates in the cell. You should qualify your findings by saying that you can't rule out off-target impacts with such a general cellular modulator as 1,6-HD, unless your controls address this and I missed it.

Response: We agree that 1,6-HD might also affect other molecules in the parasites. Because the signals (green) were from GFP-tagged UBP1, we only observed the changes in cytoplasmic distribution for the GFP-tagged molecule. We have added: “However, we cannot rule out that 1,6-HD may also have off-target effects on other molecules in the cell”. **Line 230-231.**

Line 216 - I approve of the cautious language in this sentence. Your work supports some important hypotheses, but these remain to be proven. The self-association idea is important, and I hope further work on this phenomenon in UBP1 is planned.

Response: Yes, more work can be done to investigate this phenomenon in UBP1.

Line 237 - I suggest you replace the words "than those of" with "compared to the levels of these peptides in the" ...

Response: Changed as suggested.

Lines 253-256 - Can you explain the extra (smaller) K13 bands in the NSR-NT lane of fig 5f?

Response: We do not have a definite explanation for the observation. One possibility is that even though NSS and NSR are closely related, the other differences in the parasite genomes and proteomes between the two parasites might affect K13 ubiquitination.

Line 256 - for the crt ubiquitination experiment, is it possible that crt ubiquitination occurs during a narrow time-window in the parasite? Is it possible that the parasite preparation used here did not encompass all of the intra-erythrocytic life-cycle?

Response: Yes, these are all the possibilities. CRT could be ubiquitinated early (ring stage) and is degraded by proteasome complex. We did not detect any ubiquitinated peptides from CRT based on our Ub-modified proteome analysis.

Lines 267-269 - Although it is consistent across Fig 6a and 6b, it does not appear that the change in % MDR1 localisation between the two -NT parasite lines after MQ administration is significant. Did you apply a statistical test here? My instinct would be to moderate or remove this statement of the impact of MQ pressure on this localisation measure, unless you have statistical confirmation.

Response: Thank you for pointing it out. We now performed statistical tests on the counts of cells having different MDR1 localizations within the NT parasites. The percentages of cells having PPM localized MDR1 were significantly increased after MFQ selection. One possible explanation for this observation was that the small number of NT parasites (<5%) expressing the majority of MDR1 on the DV membrane were killed after drug treatment. Now we have added the results of statistical tests on the plots (Fig. 6a and 6b).

Lines 277-283 - Please put in present tense. These sentences are a little clumsy and need improving to provide better clarity. When you write "suggesting similar temporal expression of the two proteins" do you really mean to say "suggesting that adding a second tag did not alter the temporal expression of UBPI"?

Response: Sorry for the confusion. No, we simply mean that the results from the double-tagged parasites confirmed the observations using single-tagged parasites. We have added: “, e.g., results from DTPs confirmed that both UBPI and MDR1 were expressed in blood stages...” to clarify. **Line 298.**

Line 289 - I suggest this subtitle should be "Variants of UBPI display reduced Fluo-4 ..."

Response: Changed as suggested.

Line 308 - first word "were" should be "was"

Response: Changed.

Line 309-311 - Is it better to say "These results suggest that mis-located MDR1 on the plasma membrane can mediate efflux of MFQ from parasite cytoplasm ... "

Response: Changed as suggested.

Line 322 - I suggest replace "confirm that" with "provide evidence that"

Response: Changed as suggested.

DISCUSSION

*Line 362 - delete "recently reported" as this *aat1* gene was first flagged in *P. chabaudi* by Hunt and colleagues in a publication in 2012. You can retain the Ngwa et al reference however - it is appropriate.*

Response: Corrected, thanks. Now we also cite the original paper by Modrzynska, et al. (doi: 10.1186/1471-2164-13-106).

Line 371 - The statement is too strong. I suggest replacing "may also exist" with "could emerge"

Response: Changed as suggested.

FIGURE LEGENDS

The Figures are mostly excellent, but the legends lack sufficient information and precisions in some cases. Please review these carefully.

Line 616 - in westerns, please name the antibody specificity in each case. For 3e please explain what the y-axis units are - "ubiquitinated ratio" - I can see this is normalised to the level in the IP parasites, but please spell this out the first time".

Lines 618 and 620 - you should name the technique (western blot I assume) each time if there is any ambiguity.

Response: We have rewritten the legends for Fig. 3e-3g: **“(e-g) Left panels, Western blot showing protein ubiquitination in asexual blood stages of NSS, NSR, and allelic exchanged parasites. Ub, anti-ubiquitin antibodies; Bip, anti-Bip antibodies as loading control. Right panels, the bar graphs showing quantitative signals; scanned protein signals were normalized to that of Bip and plotted. Means \pm SD of four independent experiments in (e-g); two-tailed t-test.**

(e) Global protein ubiquitination in asexual blood stages of NSS and NSR parasites.

(f) Global protein ubiquitination in asexual blood stages of allelic-exchanged parasites in NSS background.

(g) Global protein ubiquitination in asexual blood stages of allelic-exchanged parasites in NSR background”.

Line 648 - please add the phrase ""from TEM images such as those in (h)" [NB it is not (g)]... this should come after the word "puncta".

Response: Changed as suggested: “(i) Diameter measurements of UBPI-Apex2 puncta from TEM images such as those in (h)”.

Line 658 - Fix this, for example: "Proteins with the most regulated ubiquitination sites (≥ 3) in NSR compared to NSS parasites;" ...

Response: ‘ubiquitination’ added.

Line 661 - specify the tag used for IP and the Ab for the pull-down (I guess it is ubiquitin and anti-ubiquitin but it still needs to be stated).

Line 663 - specify the Ab used for the immunoblotting.

Response: We have added the specific antibodies: “...followed by IP (anti-tag antibodies) and immunoblotting (IB, anti-ubiquitin antibodies) as indicated; Bip, anti-Bip antibodies as protein loading control”.

Colin Sutherland
LSHTM
London

Reviewer #2 (Remarks to the Author):

This is a comprehensive and exceedingly thoroughly-controlled study on the function of the Plasmodium UBPI deubiquitinating enzyme.

UBPI has been long associated with parasite drug resistance and point mutations within its sequence have become markers for artemisinin resistance in field isolates. To date, no substrates or biological function have been attributed to this enzyme beyond it being localised to the parasite cytosome. As such, this study will undoubtedly be of considerable interest to the wider malaria community and will be important in influencing further work into how drug resistance arises.

The authors identified two point mutations that spontaneously arose by cycling parasites under drug pressure. These mutations together conferred a fitness defect (interesting in itself

considering other point mutant do not have this defect), an endocytic defect as measure by lower haemozoinb in the food vacuole, higher systemic ubiquitination and mislocalisation of the UBPI protein.

Ubiquitinome analysis revealed MDR1 was differentially ubiquitinated along with a set of Kelch13 interacting proteins. The authors demonstrated with a set of elegant approaches and convincing experiments that in the absence of UBPI-mediated deubiquitination, MDR1 mislocalises from the FV to the PM where it serves as a drug efflux pump, thereby conferring drug resistance to the parasite.

I find the data convincing, well-controlled, and of great significance to malaria cell biology. As such, I only have minor comments and corrections to suggest.

Thank you for the positive comments!

Minor corrections/comments:

The penicillin binding domain is novel but no explanation is given as to what it might be doing. It would be helpful if the authors discussed it more thoroughly.

Response: Again, thank you for pointing this out. Using the Swiss-Model server, we previously identified a UBPI amino acid region harboring the I1560N mutation that has some homology with the motifs III-IV of the glycosyltransferase (GTase) domain in bacterial Penicillin-binding proteins (PBPs) (**Fig. S4a**). The PBPs are bifunctional proteins containing GTase activity and transpeptidase (TPase) activity for peptidoglycan synthesis in bacteria including *Escherichia coli* and *Aquifex aeolicus* (doi:10.1016/j.jmb.2008.08.020). However, the UBPI region shared only ~30% similarity with the GTase domain of the bacterial PBPs. Additionally, the UBPI 'KKKIMDEILNTL' motif (motif IV of bacterial PBP1) is also homologous to an ATP-binding cassette sub-family G member 1-like isoform X1 of *Belonocnema kinseyi* (**Fig. S4b**) and a DNA repair protein of *Campylobacter concisus* (**Fig. S4c**). We, therefore, decided to replace the "PBP domain" with "conserved central region (CCR) domain" in the text due to the lack of further functional characterization of the domain.

We have modified the sentence to read: "The NT mutations were located within a conserved central region (CCR) and ubiquitin carboxyl-terminal hydrolase (UCH) domains (Supplementary Fig. 3), respectively. The CCR has some homology with the motifs III-IV of the glycosyltransferase (GTase) domain of bacterial penicillin-binding proteins (PBPs) (Supplementary Fig. 4a), as predicted using the Swiss-Model automated server (<https://swissmodel.expasy.org/>). The PBPs are bifunctional proteins containing GTase activity and transpeptidase (TPase) activity for peptidoglycan synthesis in bacteria including *Escherichia coli* and *Aquifex aeolicus*²⁵. However, the CCR region shared only ~30%

similarity with the motifs III-IV in GTase domain of the bacterial PBPs. Additionally, the UBP1 ‘KKKIMDEILNTL’ motif (motif IV of bacterial PBP1) is also homologous to an ATP-binding cassette sub-family G member 1-like isoform X1 of *Belonocnema kinseyi* (Supplementary Fig. 4b) and a DNA repair protein of *Campylobacter concisus* (Supplementary Fig. 4c). The function of the UBP1 CCR is uncertain, and further investigation is required. To our knowledge, the IP→NT substitutions were identified for the first time, and the CCR domain has not been characterized in malaria parasites previously.”

Line 115-130

Supplementary Fig. 4. Sequence alignment to bacterial Penicillin-Binding Proteins (PBPs) and a conserved UBP1 motif. (a), Partial sequence homology of *Plasmodium* UBP1 amino acid sequence with bacterial Penicillin-Binding Proteins (PBPs) was predicted using the Swiss-Model automated server (<https://swissmodel.expasy.org/>) and aligned using Jalview (<https://www.jalview.org/>). (b), Sequence alignment and partial homology to an ATP-binding cassette using BLAST searches. (c), Sequence alignment to a motif in DNA repair protein using BLAST searches.

Some of the IFA figures are distorted- these need to be amended to publication (example figure 4F)

Response: For clarity, we replaced the Nuclei/GFP panels with DIC/Merge panels in Fig 4f.

Figure 6a/b: the authors show that MDR1 is largely mislocalised to the PM in the presence of the UBPI double mutant but that this effect is even more pronounced under drug pressure. What do they propose the mechanisms of this additional drug-induced effect is?

Response: We do not know why there were still a small number of parasites with DVM expression. One possible explanation is that a small number (less than 5%) of parasites having the NT haplotype that do not translocate MDR1 from the DVM to the PMM efficiently are killed after drug selection. We have added: “One possibility for this observation was that MDR1 had not been adequately ubiquitinated in the small number of parasites. Alternatively, some parasites might revert to IP/NP/IT genotypes and express some MDR1 on DVM”. Now we also added the results of statistical tests on the plots as suggested by reviewer 1 (Fig. 6a and 6b). Line 287-290.

The data presented pertaining to the self-association of UBPI being necessary for proper localisation was a bit over interpreted. Are there other examples of DUBs in other biological systems where oligomerisation is necessary for activity? Discussing such cases would strengthen the arguments.

Response: Thank you for asking the question! Yes, there are many examples showing that oligomerization of DUBs can affect their localization and activities (doi: 10.1016/j.jbc.2022.102198). For example, human USP25 is assembled into a homotetrameric quaternary complex that inhibits its enzymatic activity (doi: 10.1038/s41467-018-07510-5).

We have added: “Indeed, oligomerization is one of the mechanisms that can regulate DUB activities³⁰. For example, human USP25 is assembled into a homotetrameric quaternary complex that inhibits its enzymatic activity³¹”. Line 233-235.

If MDR1 is indeed functioning as a drug efflux pump at the PPM, and given its prior association with DHA and artemisinin resistance, it seems strange that DHA resistance is unaffected in this study. The authors touch upon this in the discussion by saying different point mutation in UBPI may give rise to different drug resistance profiles. Would this suggest

that MDR1 is not actually involved in DHA/ART resistance?

Response: This is an interesting question. As we speculated in response to reviewer 1's question (see below), the issue may be related to parasite stages (ring vs trophozoite and schizont). DHA 'resistance' is mostly characterized by better survival of the ring measured by ring survival assay (RSA). *P. falciparum* parasites having an increased copy number of MDR1 and/or K13 mutations are often called DHA resistant after RSA measurements; but they are not resistant to DHA using the classical criteria of drug resistance (WHO 3-4 levels of resistance). In other words, the 'DHA resistant' parasites still have low IC₅₀ values (<10 nM) in *in vitro* 72 drug survival assays. In our *in vivo* drug assay, we treated the mice with four doses over four days. During this period, DHA will kill the trophozoite and schizont stages due to multiple rounds of drug treatments. Therefore, the UBP mutations do not affect parasite survival after four days of DHA treatment. We may see a difference in the ring stage; however, it is difficult to isolate the ring stage of *P. yoelii*, and we cannot culture the parasites for as long term as we can for *P. falciparum*. In other words, using a different assay (short-term drug treatment), we may see a difference in parasite response to ART between the WT and mutant parasites.

To address this issue, we have added: "Our *in vivo* drug assay showed no effect of the UPB1 IP→NT substitutions on parasite growth after ART treatment (Fig. 2c), which does not conflict with the 'ART resistant' phenotype based on ring survival assay (RSA) or delayed parasite clearance *in vivo*. The RSA is a phenotype only for a brief window (~6h) of intra-erythrocytic development, whereas in our *in vivo* drug assays, the parasites were treated 4 doses over several days" in the Discussion. **Line 422-427.**

Reviewer #3 (Remarks to the Author):

The manuscript on mutant ubp1 relocating mdr1 in the mouse model is a tour de force with well crafted studies laid out in a logical manner with a thorough approach to nailing the evidence.

Thank you for the positive comments!

*The title should note genus and species. Many drug resistant mechanisms differ between *P. vivax* or *falciparum* and the murine *Plasmodium*. The last line of the abstract also needs to contain the words "in a mouse model" after substrate transport direction, in a mouse model, providing...*

Response: We have added '*yoelii*' to the title, and 'in a mouse model' was added to the abstract as suggested.

While the authors noted in line 103 on page 5 no mutation or copy number variation in orthologues of *Pfcr1*, *pfmdr1*, *pfmdr2* and *pfk13* they left out *plasmepsin*. This copy number or mutation should be looked for in existing data as in *P. falciparum* they are often opposites in number of copies with piperquine resistance with increase CNV in *plasmepsin* having low CNV in *pfmdr1* and vice versa.

Response: Thank you for the suggestion. We have checked *plasmepsin* genes including *plasmepsin 2* and *plasmepsin 3* that are associated with piperquine resistance, no mutation or copy number variation was found in both *plasmepsin* genes between the isogenic lines NSS and NSR.

We have added the information to the sentence: “No mutation or copy number variation was found in the orthologues of *Pfcr1*, *Pfmdr1*, *Pfmdr2*, *Pfk13*, *plasmepsin 2*, and *plasmepsin 3* that have been implicated in various drug resistances”. Line 104-106.

The results section on page 7 line 150 to 153 need to explicitly note if resistance to both mefloquine and piperquine tracked together or diverged amongst the mutants in NSS and NSR. This reviewer favors identical linear or log x axis for figure 2C as the DHA decreasing by 10 mg/kg is confused with mfq or ppq with x axis 40 or 60. I would add a table of mg/kg for IC₅₀, IC₉₀ and IC₉₉ for isolates and drugs to clarify the figure 2 C phenotypes by drugs.

Response: We have added: ‘simultaneous’ to read: “These findings indicate that double mutations (1560N plus 2874T) are required for simultaneous resistance to high levels of MFQ, LUM, and PPQ”. Line 165.

We have plotted the x-axis in log₂ scale, which provides more uniform spacing.

We have calculated the IC₅₀ and IC₉₀ for the MFQ, LUM, PPQ, and DHA for the IP/NP/IT parasites as well as DHA for the NT parasites and presented the data in the new **Supplemental Table S3**. We were not able to calculate the IC₅₀ and IC₉₀ values for the NT parasites due to their survival under high drug concentrations. Similarly, IC₅₀ and IC₉₀ values could not be calculated for CQ and QN due to the relatively low drug dosages used.

Plasmepsins can be brought into the discussion

Response: In the Discussion, we now include *plasmepsins*: “Similarly, the V2697F and V2728F substitutions may also regulate the ubiquitination, trafficking, and function of other proteins such as CRT, K13, *plasmepsins*, or the amino acid transporter AAT1^{47,48} to influence parasite drug responses. There are approximately 30 predicted DUBs in the *Plasmodium* genome^{49,50}. Although UBP1 is shown to deubiquitinate many parasite proteins, other DUBs may also affect parasite protein ubiquitination and responses to drugs. Additionally, mutations in PfCRT and copy number variation of *plasmepsins 2/3* have also been linked to PPQ resistance^{51,52}. The relationships of CRT, *plasmepsins*, MDR1, and UBP1 (or other DUBs) and their roles in drug resistance required further investigations”. Line 384-392.

Finally, we would like to thank the reviewers for the positive and constructive comments/suggestions that have greatly improved our manuscript.